# Nearly Optimal Sample Complexity for Learning with Label Proportions

**Robert Busa-Fekete** [* 1] **Travis Dick** [* 1] **Claudio Gentile** [* 1] **Haim Kaplan** [* 1 2] **Tomer Koren** [* 1 2] **Uri Stemmer** [* 1 2]

## Abstract

We investigate Learning from Label Proportions (LLP), a partial information setting where examples in a training set are grouped into bags, and only aggregate label values in each bag are available. Despite the partial observability, the goal is still to achieve small regret at the level of individual examples. We give results on the sample complexity of LLP under square loss, showing that our sample complexity is essentially optimal. From an algorithmic viewpoint, we rely on carefully designed variants of Empirical Risk Minimization, and Stochastic Gradient Descent algorithms, combined with ad hoc variance reduction techniques. On one hand, our theoretical results improve in important ways on the existing literature on LLP, specifically in the way the sample complexity depends on the bag size. On the other hand, we validate our algorithmic solutions on several datasets, demonstrating improved empirical performance (better accuracy for less samples) against recent baselines.

## 1. Introduction

Learning with Label Proportions (LLP) is a semi-supervised learning setting where the learning algorithm only observes label values of a training set at the resolution of groups of feature vectors. Specifically, the training set is partitioned into collections of unlabeled feature vectors, called *bags* in the literature, and only the proportion of positive labels within each bag are observed, instead of the individual labels. The motivation comes from a number of practical scenarios where the access to individual labels is either costly or impossible to achieve, or made available to a learner at an aggregate level because of privacy-preserving concerns.

Among these practical scenarios, it is worth mentioning

e-commerce, medical databases (Patrini et al., 2014), high energy physics (Dery et al., 2018), election prediction (Sun et al., 2017), medical image analysis (Bortsova et al., 2018), remote sensing (Ding et al., 2017).

Though early contributions to LLP date back to the mid 2000s (e.g., (de Freitas & Kuck, 2005; Musicant et al., 2007; Quadrianto et al., 2008)), it is only more recently that we have seen a general resurgence of interest in this problem (e.g., (Dulac-Arnold et al., 2019; Lu et al., 2019; Scott & Zhang, 2020; Saket, 2021; 2022; Tsai & Lin, 2020; Lu et al., 2021; Zhang et al., 2022; Busa-Fekete et al., 2023; Li et al., 2024; Havaldar et al., 2024)), often related to the desire of preserving the privacy of user data/activity in data-intensive online businesses.

Relevant practical scenarios where LLP has started to play a pivotal role relate to the ad conversion reporting systems proposed by Apple (SKAN, (app, 2025)) and Google Chrome and Android (Privacy Sandbox (chr, 2021)), where user conversions (like user purchases) are only available at the aggregate level (e.g., by ad campaign), and training a conversion model is thus forced to leverage such aggregate signals as much as possible. Still, the models trained this way are required to perform well at the level of *individual* conversions.

This paper is inspired by two recently published works in the LLP literature (Busa-Fekete et al., 2023; Li et al., 2024), where regret guarantees are given under standard statistical learning assumptions for randomly drawn (and non-overlapping) bags of a given size. In the first paper, the authors describe a general debiasing technique that applies to any bounded loss function (or loss gradient) to turn aggregate label signal into an individual label signal at the cost of an inflated variance of the resulting estimator. This allows the authors to seamlessly adapt their technique to standard learning methods like Empirical Risk Minimization (ERM) and Stochastic Gradient Descent (SGD). In particular, the authors give regret guarantees in general non-realizable scenarios showing that when the learner deals with bags of size $k$, the sample complexity of ERM and SGD with label proportions increases by a factor of roughly $k^2$.

In (Li et al., 2024), the authors improve upon (Busa-Fekete et al., 2023) when the loss function is the square loss, in that their bounds yield fast rates of convergence in realizable

---

[*]Equal contribution [1]Google Research, New York, USA [2]Tel Aviv University, Israel. Correspondence to: Claudio Gentile <cgentile@google.com>, Travis Dick <tdick@google.com>.

*Proceedings of the 42nd International Conference on Machine Learning*, Vancouver, Canada. PMLR 267, 2025. Copyright 2025 by the author(s).

settings (unlike (Busa-Fekete et al., 2023)). Yet, this is achieved at the cost of a widely sub-optimal dependence on the bag size $k$. In this paper, we leverage variance reduction techniques to achieve essentially optimal (up to log factors) sample complexity guarantees for LLP under square loss for both ERM and SGD. For instance, a quick comparison to (Busa-Fekete et al., 2023; Li et al., 2024) for square loss reveals the following. Call $\beta > 0$ the target regret guarantee. Then our sample complexity bounds are of the form $\frac{k}{\beta^2}$ in the non-realizable case, and $\frac{k}{\beta}$ in the realizable case. In contrast, (Busa-Fekete et al., 2023) only contains non-realizable (i.e., slow rate) guarantees, which are of the form $\frac{k^2}{\beta^2}$, while the realizable (fast rates) guarantees from (Li et al., 2024) are of the form $\frac{k^3}{\beta}$. We also show that in the realizable case the sample bound $\frac{k}{\beta}$ is the best one can hope for.[1]

Moreover, we perform a thorough set of experiments, where we compare our algorithm to the folklore *proportion matching* algorithm, which was reported to be a strong baseline in (Busa-Fekete et al., 2023), to the debiasing method from (Li et al., 2024), and to a number of alternative methods available in the LLP literature. The general goal of these experiments was to validate our theory on improved sample complexity guarantees. On diverse combinations of datasets and learning models, our experiments do indeed reveal the superior performance of our variance reduction methods over the tested baselines. The empirical performance difference is particularly stunning when the training procedure is restricted to only a few training epochs, thereby showcasing a substantial enhancement in convergence speed.

### 1.1. Related work

Initial interest in LLP dates back to at least (de Freitas & Kuck, 2005; Musicant et al., 2007; Quadrianto et al., 2008; Rueping, 2010; Stolpe & Morik, 2011; Yu et al., 2013; Patrini et al., 2014). In (de Freitas & Kuck, 2005) the authors propose a hierarchical model generating labels according to the given label proportions, and formulates an MCMC-based inference scheme which, unfortunately, does not scale to large datasets. Musicant et al. (2007) stress that standard supervised learning algorithms (like SVM and $k$-Nearest Neighbors) can be adapted to LLP by a reformulation of their objective functions, but no experiments are reported in that paper on classification tasks. Quadrianto et al. (2008) give an algorithm to learn an exponential generative model, which was further generalized in (Patrini et al., 2014). Both papers, however, make very strong modeling assumptions, like conditional exponential models, which are often inadequate for Deep Neural Network (DNN) and more modern

machine learning architectures. Rueping (2010); Yu et al. (2013) propose an adaptation of SVMs to LLP. In particular, Yu et al. (2013) propose an SVM algorithm for regression which optimizes the SVM loss by matching available label proportions (this general approach is called *proportion matching* in (Busa-Fekete et al., 2023)). Yet, their approach turns out to be restricted to linear models in some feature space. Similar limitations are contained in other SVM-based papers on LLP, like (Qi et al., 2017; Shi et al., 2019). Being very natural, the proportion matching idea was extended to other classifiers. For instance, Li & Taylor (2015) extends the formulation to CNNs with a generative model where the maximum likelihood estimator is computed via Expectation Maximization. Yet, this turns out not to be scalable even to moderately large DNN architectures. Liu et al. (2019) experimentally investigate multiclass LLP via Generative Adversarial Networks. Another very recent experimental paper is (Havaldar et al., 2024), where the authors adopt an iterative process with two phases, in which a Gibbs distribution is first defined over labels that factors in the feature vectors so as to force nearby vector to have similar labels, and then use Belief Propagation to obtain pseudo-labels that produce embedding refinements.

Among the more theoretically-oriented papers on LLP are (Saket, 2021; 2022; Brahmbhatt et al., 2023; Lu et al., 2019; 2021; Scott & Zhang, 2020; Zhang et al., 2022), as well as the already mentioned (Busa-Fekete et al., 2023; Li et al., 2024). In (Saket, 2021; 2022; Brahmbhatt et al., 2023) the authors are essentially restricting to linear-threshold functions and in some cases rely on the fact that bags can be overlapping. In contrast, we are following here the learning setting of Busa-Fekete et al. (2023); Li et al. (2024), and working with non-overlapping i.i.d. bags with more general model classes.

In (Lu et al., 2019; 2021) the authors tackle the problem of learning from multiple unlabeled datasets, which is similar to LLP. The authors propose a debiasing method for the loss function, and prove consistency results (which are similar in spirit to those contained here, as well as those in (Busa-Fekete et al., 2023; Li et al., 2024)), but they do so by imposing further restrictions, like the separation of the class priors over bags. This can only be done by enabling access to the class conditional distributions. In contrast, the bags proposed in our setup are drawn i.i.d. from the same prior distribution, a setting where the algorithms proposed in (Lu et al., 2019; 2021) would fail.

Scott & Zhang (2020); Zhang et al. (2022) reduce LLP to learning with label noise, in that bags are paired, and each pair is seen as a problem of learning with label noise, with label proportions being treated akin to label flipping probabilities. Yet, as in (Lu et al., 2019), the bag pairing heavily relies on the statistical diversity of the bags, which

---

[1]A similar statement can also be extracted from (Li et al., 2024). Their lower bound has a wider scope than ours, but at the cost of extra $\log k$ factors.

we explicitly rule out. The kind of population loss they are interested in (balanced risk) is also a bit different from the one we deal with here.

For the sake of fair comparison, in our experiments (Section 6) we followed prior experimental setups, like the one in (Busa-Fekete et al., 2023), and chose to compare only to methods with similar goals and settings as ours, like the folklore proportion matching method (e.g., (Yu et al., 2013)), the EasyLLP method by Busa-Fekete et al. (2023), the debiased square loss method proposed by Li et al. (2024), and the label generation approach by Dulac-Arnold et al. (2019) (binary version thereof).

## 2. Preliminaries

For a natural number $n$, let $[n] = \{i \in \mathbb{N} : i \leq n\}$. Let $\mathcal{X}$ be a feature (or instance) space, and $\mathcal{Y}$ be a binary $\mathcal{Y} = \{0, 1\}$ label space. Let $\mathcal{D}$ be a joint distribution on $\mathcal{X} \times \mathcal{Y}$, encoding the correlation between the input features and the labels. The marginal distribution over $\mathcal{X}$ will be denoted by $\mathcal{D}_{\mathcal{X}}$, and the conditional distribution of $y$ given $x$ will be denoted by $\mathcal{D}_{\mathcal{Y}|x}$. In particular, we denote by $\eta(x) = \mathbb{P}_{y \sim \mathcal{D}_{\mathcal{Y}|x}}(y = 1|x) = \mathbb{E}_{y \sim \mathcal{D}_{\mathcal{Y}|x}}[y|x]$ the probability of drawing label 1 conditioned on feature vector $x$. When clear from the surrounding context, we shall remove the subscripts from probabilities and expectations and, e.g., abbreviate $\mathbb{E}_{y \sim \mathcal{D}_{\mathcal{Y}|x}}[y|x]$ by $\mathbb{E}[y|x]$. A dataset $S = (x_1, y_1), \ldots, (x_n, y_n)$ is a sequence of pairs $(x_i, y_i)$, each one drawn i.i.d. from $\mathcal{D}$.

In our LLP scenario, the dataset $S$ is partitioned uniformly at random into *bags* of a given size $k$, where only feature vectors and the fraction of positive labels in each bag are available for learning, that is,

$$S = \underbrace{((x_{1,1}, \ldots, x_{1,k}), \alpha_1)}_{(\mathcal{B}_1, \alpha_1)}, \ldots, \underbrace{((x_{m,1}, \ldots, x_{m,k}), \alpha_m)}_{(\mathcal{B}_m, \alpha_m)} .$$

(1)

with $n = mk$, where $\mathcal{B}_j = \{x_{j,i} : i \in [k]\}$, $\alpha_j = \frac{1}{k} \sum_{i=1}^{k} y_{j,i}$ is the label proportion (fraction of labels "1") in the $j$-th bag, and all the involved samples $(x_{j,i}, y_{j,i})$ are drawn i.i.d. from $\mathcal{D}$. Thus, the learning algorithm gets information about the $n$ labels $y_{j,i}$ of the $n$ instances $x_{j,i}$ from dataset $S$ only in the aggregate form determined by the $m$ label proportions $\alpha_j$. Yet, note that the feature vectors $x_{j,i}$ are individually observed. Also, note that, for a given sample size $n = mk$, the bigger $k$ the smaller the overall amount of label information the algorithm receives.

As is standard in statistical learning, we are given a hypothesis space $\mathcal{H} = \{h : \mathcal{X} \to [0, 1]\}$ of functions $h$ mapping $\mathcal{X}$ to $[0, 1]$, where $h(x)$ can be interpreted as the probability that $y = 1$ given $x$ according to model $h$, and a loss function $\ell : [0, 1] \times \mathcal{Y} \to \mathbb{R}^+$. In LLP the learner has access to $S$ in the form (1) above, and tries to find a hy-

pothesis $\widehat{h} \in \mathcal{H}$ with the smallest *population loss* (or *risk*) $\mathcal{L}(h) = \mathbb{E}_{(x,y) \sim \mathcal{D}}[\ell(h(x), y)]$ with high probability over the random draw of $S$. In order not to clutter the paper with too many mathematical details, we shall assume henceforth that the hypothesis space $\mathcal{H}$ is finite. The theoretical analyses contained in the paper can be easily lifted to infinite hypothesis spaces via standard tools in empirical process theory (e.g., (Bartlett et al., 2005; Boucheron et al., 2012; Vershynin, 2018)).

Given $\mathcal{H}$ and $\mathcal{D}$, the *excess risk* (or *regret*) $\mathcal{R}(\widehat{h})$ of $\widehat{h}$ is $\mathcal{R}(\widehat{h}) = \mathcal{L}(\widehat{h}) - \mathcal{L}(\widehat{h}^\star)$, where $\widehat{h}^\star = \arg\min_{h \in \mathcal{H}} \mathcal{L}(h)$ is the model in $\mathcal{H}$ having the smallest risk (sometimes called *best-in-class* model). We say that we are in a *realizable* setting when the Bayes-optimal predictor $h^\star = \arg\min_h \mathcal{L}(h)$, the minimum being taken over all possible (measurable) functions, belongs to $\mathcal{H}$, and in a *non-realizable* setting, otherwise. Clearly enough, in the realizable setting, $\widehat{h}^\star = h^\star$.

Our goal is to design and analyze algorithms that compute $\widehat{h} \in \mathcal{H}$ with small $\mathcal{R}(\widehat{h})$ in both realizable and non-realizable settings. In particular, for a given bag size $k$, our goal is to minimize the number of bags $m$ (the sample complexity is then $n = mk$) required for a particular regret guarantee. We will obtain tight bounds on the sample complexity specifically for the *square loss* case, $\ell(h(x), y) = (y - h(x))^2$.

In a sense, if we view label aggregation as a form of (label) privatization mechanism (which is not differentially private, though), this investigation aims at striking the best possible trade-offs between utility (that is, the regret $\mathcal{R}(\widehat{h})$ as a function of the sample size $n$) and privacy in the form of the label aggregation level $k$.

As in recent investigations on LLP (Busa-Fekete et al., 2023; Li et al., 2024), we cover two kinds of standard learning algorithms: Empirical Risk Minimization (ERM), and Stochastic Gradient Descent (SGD), and show that we can improve in various ways on the results of both papers.

## 3. Warmup: A realizable Setting with Two Functions

As a warmup, we consider the simplest possible scenario of LLP, a realizable setting where the function space $\mathcal{H}$ has only two models $\mathcal{H} = \{h_1, h_2\}$, with either $\eta(\cdot) = h_1(\cdot)$ or $\eta(\cdot) = h_2(\cdot)$. Though the argument we are about to present here clearly applies to more general loss functions, consider for definiteness the square loss. It can be easily seen that in this case the Bayes optimal predictor $h^\star(\cdot)$ coincides with $\eta(\cdot)$.

Suppose the two models are significantly different from one another, in the sense that $|\mathcal{L}(h_1) - \mathcal{L}(h_2)| = \beta > 0$. In

this case, if we want achieve a regret smaller than $\beta$, we are forced to identify $h^\star$ exactly. We will see momentarily that this simple problem already poses significant challenges if we want to prove optimal rates of convergence as a function of both the number of bags $m$, and the bag size $k$.

Our algorithm for computing $\widehat{h}$ is presented as Algorithm 1. For bag $(\mathcal{B}_j, \alpha_j)$, and hypothesis $h_s$, for $s \in \{1, 2\}$, set for brevity

$$\mathbb{E}_{j,s} = \mathbb{E}[k\alpha_j \,|\, \mathcal{B}_j] = \sum_{x \in \mathcal{B}_j} h_s(x) \,.$$

Denote by $\{\cdot\}$ the indicator of the predicate at argument, define $\mu_j = \frac{1}{2}(\mathbb{E}_{j,1} + \mathbb{E}_{j,2})$, for $j \in [m]$, and consider the statistics

$$A_1 = \sum_{j=1}^m (k\alpha_j - \mathbb{E}_{j,1}) \qquad A_2 = \sum_{j=1}^m (k\alpha_j - \mathbb{E}_{j,2})$$

and

$$Q = \sum_{j=1}^m \Big( \{\mathbb{E}_{j,1} \le \mathbb{E}_{j,2}\}(k\alpha_j - \mu_j)$$
$$+ \{\mathbb{E}_{j,1} > \mathbb{E}_{j,2}\}(\mu_j - k\alpha_j) \Big) \,.$$

Define the bias $\Delta = \mathbb{E}[h_1(x) - h_2(x)] \in [-1, 1]$, and note that, since the underlying loss is the square loss, the separation parameter $\beta$ can also be written as $\beta = \mathbb{E}[(h_1(x) - h_2(x))^2]$. We assume here for simplicity that both $\Delta$ and $\beta$ are known or easy to estimate. In fact, both quantities can be estimated without labels, and we can do so with an accuracy which is higher than the one allowed by the $m$ aggregate label signals (Section B.2 contains an example of these estimates applied to the more involved scenario of Section 4).

Intuitively, when $|\Delta|$ is large, the models $h_1$ and $h_2$ are distinguishable based only on the total number of positive predictions they make. In this case, the algorithm predicts based on $A_1$ and $A_2$, which compare the observed number of positive labels to the number expected under $h_1$ and $h_2$, respectively. However, when $|\Delta|$ is small, the algorithm needs to check for agreement between $h_1$ and $h_2$ and the label proportion on each bag, which we do through $Q$.

The following theorem[2] shows that a sample complexity $n = O(k/\beta)$ (without hidden log factors) is both necessary and sufficient. The result improves on (Li et al., 2024) by shaving off the $\log k$ factors appearing in both the upper and the lower bounds analysis therein (see Theorem 1 and Theorem 8 in (Li et al., 2024)).

**Theorem 3.1.** *Let $\mathcal{H} = \{h_1, h_2\}$ be a hypothesis class where $h^\star$ is either $h_1$ or $h_2$, $\mathcal{Z} = \mathcal{X} \times \mathcal{Y}$ be the domain*

---

[2]Full proofs are given in the appendix.

---

**Algorithm 1** Algorithm for the two function realizable case.

**Input:** Sample $S$ made up of $m \ge 1$ bags, each of size $k \ge 1$, bias $\Delta \in [-1, 1]$, separation parameter $\beta \in (0, 1]$.

- **If** $|\Delta| \ge \sqrt{\frac{\beta}{2k}}$ **then:**
    - **If** $|A_1(S)| < |A_2(S)|$ **then:** $\widehat{h} = h_1$
    - **Else:** $\widehat{h} = h_2$

- **Else** $\widehat{h} = \begin{cases} h_2 & \text{if } Q(S) \ge 0 \\ h_1 & \text{if } Q(S) < 0 \end{cases}$

**Output:** $\widehat{h}$

---

where a probability distribution $\mathcal{D}$ is defined,

$$\beta = \mathbb{E}[(h_1(x) - h_2(x))^2] \,,$$

and

$$\Delta = \mathbb{E}[h_1(x) - h_2(x)] \in [-1, 1] \,.$$

Let Algorithm 1 be fed with an i.i.d. sample of size $n$ drawn according to $\mathcal{D}$, with $m$ bags each of size $k$. Then, if $k = \Omega(1/\beta)$, with probability at least $1 - \delta$ Algorithm 1 outputs $\widehat{h} = h^\star$, provided

$$n = O\left( \min\left\{ \frac{1}{\Delta^2}, \frac{k}{\beta} \right\} \log \frac{1}{\delta} \right) \,.$$

Moreover, when $\Delta = 0$ (which is the hardest case), a specific setting of $\mathcal{H}$ and $\mathcal{D}$ exists for which no algorithm can be correct with probability at least $1 - \delta$, with $\delta < 1/2$, unless

$$n = \Omega\left( \frac{k(1 - \mathcal{E}(\delta))}{\beta} \right) \,,$$

where $\mathcal{E}(\cdot)$ is the binary entropy function $\mathcal{E}(x) = -x \log_2 x - (1-x) \log_2(1-x)$. In other words, when $\Delta = 0$ the sample size $n$ must be linear in $k$.

Algorithm 1 can immediately be extended to the case of more than two hypotheses, just by implementing an elimination tournament via pairwise comparisons. We do not know, however, how to make it work in non-realizable settings.

The reader should observe that, being the decision rule $Q$ a *pairwise* statistic, it cannot be expressed as the difference of two separate scoring functions (one for $h_1$ and the other for $h_2$). Hence Algorithm 1, though sample optimal, does not easily lend itself to extensions of practical relevance. We now turn our attention to more viable solutions.

## 4. Empirical Risk Minimization

Our proposed Empirical Risk Minimization (ERM) algorithm for square loss operates on a bag level loss that can be

seen as a robust (or clipped) bag-level square loss. Specifically, for bag $z_j = (\mathcal{B}_j, \alpha_j)$ and function $h \in \mathcal{H}$, we set

$$\mathbb{E}_j(h) = \sum_{i=1}^{k} h(x_{j,i}) , \quad \widetilde{\mathbb{E}}_j(h) = \mathbb{E}_j(h) - k\mathbb{E}[h(x)] .$$

Then we define the *clipped* bag-level square loss

$$\ell^c(h, z_j) = \frac{1}{k}\left(k\widetilde{\alpha}_j - \widetilde{\mathbb{E}}_j(h)\right)^2 G_j(h) + \left(\mathbb{E}[h(x)] - p\right)^2 ,$$

where $\widetilde{\alpha}_j = \alpha_j - p$, $p = \mathbb{E}[h^\star(x)]$, and

$$G_j(h) = \left\{ |k\widetilde{\alpha}_j - \widetilde{\mathbb{E}}_j(h)| \leq \sqrt{8k \log \frac{2}{\theta}} \right\} ,$$

for a suitable value of parameter $\theta > 0$. (Recall that $\{\cdot\}$ denotes the indicator function for the predicate at argument.)

Again, for simplicity of presentation, we assume in the above that $\mathbb{E}[h(x)]$, for $h \in \mathcal{H}$, and $p$ are known. In fact, $\mathbb{E}[h(x)]$ can be easily estimated from *unlabeled* data, and $p$ can be easily estimated from label proportions $\alpha_j$, the main point being that both quantities can be estimated at a higher resolution than the one allowed by the aggregate label signals. Appendix B.2 contains a version of the algorithm where $\mathbb{E}[h(x)]$ and $p$ are themselves estimated from data. This is similar in spirit to the corresponding argument in (Li et al., 2024), but the analysis here is somewhat more involved, since special care has to be taken to ensure that fast rates are preserved even after clipping the square loss.

Then we define the ERM estimator

$$\widehat{h}(S) = \arg\min_{h \in \mathcal{H}} \frac{1}{m} \sum_{j=1}^{m} \ell^c(h, z_j) .$$

We have the following result.

**Theorem 4.1.** *Let $\mathcal{H}$ be a finite hypothesis space, $\mathcal{Z} = \mathcal{X} \times \mathcal{Y}$ be the domain where a probability distribution $\mathcal{D}$ is defined. Let the ERM estimator $\widehat{h}$ defined above be fed with a sample of size $n = mk$, with $m$ bags of size $k$, drawn i.i.d. according to $\mathcal{D}$, and let*

$$\gamma(\widehat{h}^\star, h^\star) = \mathbb{E}[(\widehat{h}^\star(x) - h^\star(x))^2] .$$

*Then, for all $\beta > 0$, if $\theta = \frac{\beta}{16k^2}$ in the clipping condition defining $G_j(h)$, and*

$$m = O\left( \frac{(\gamma(\widehat{h}^\star, h^\star) + \beta)\log \frac{k}{\beta}}{\beta^2} \log \frac{|\mathcal{H}_\beta|}{\delta} \right) ,$$

*we have*

$$\mathcal{R}(\widehat{h}) = \mathcal{L}(\widehat{h}) - \mathcal{L}(\widehat{h}^\star) < \beta$$

*holding with probability at least $1 - \delta$. In the above,*

$$\mathcal{H}_\beta = \left\{ h \in \mathcal{H} : \mathcal{R}(h) \geq \beta \right\} \subseteq \mathcal{H} .$$

*In particular, in the realizable case ($\gamma(\widehat{h}^\star, h^\star) = 0$), the number of bags $m$ that suffices is*

$$m = O\left( \frac{\log \frac{k}{\beta}}{\beta} \log \frac{|\mathcal{H}_\beta|}{\delta} \right) .$$

*Moreover, a similar algorithm exists, where the quantities $\mathbb{E}[h(x)]$ and $p$ are replaced by empirical estimates, which enjoys the same sample complexity guarantees as above.*

*Proof sketch.* Define the *non-clipped* version of the loss at the bag level:

$$\ell(h, z_j) = \frac{1}{k}\left(k\widetilde{\alpha}_j - \widetilde{\mathbb{E}}_j(h)\right)^2 + \left(\mathbb{E}[h(x)] - p\right)^2 , \quad (2)$$

and let $\mathcal{L}_\mathcal{B}(h)$ denote the expected value of $\ell(h, z_j)$ over the random draw of bag $z_j$. The first observation we make is that this expectation is the same as the expectation of the original square loss $(y - h(x))^2$, when $(x, y)$ is randomly drawn according to $\mathcal{D}$, i.e.,

$$\mathcal{L}_\mathcal{B}(h) = \mathcal{L}(h) . \quad (3)$$

Then the proof essentially proceeds as a bias-variance trade-off analysis. We show that the clipping operation that turns $\ell(h, z_j)$ into $\ell^c(h, z_j)$ only introduces a small bias while, at the same time, helping us reduce the variance substantially.

In fact, as is customary in fast rate analyses (e.g., (Massart, 2000; Mendelson, 2002; Bartlett et al., 2005)) the above bias-variance trade-off does not refer to the loss itself, but to the *difference* of losses $\ell^c(h, z_j) - \ell^c(\widehat{h}^\star, z_j)$, for which we can prove a bias bound of the form $k\theta$, a variance bound of the form

$$\gamma(h, \widehat{h}^\star) \log \frac{1}{\theta} + k^2\theta ,$$

and a range bound of the form $\log \frac{1}{\theta}$. In turn, for square loss,

$$\gamma(h, \widehat{h}^\star) = O\big(\gamma(\widehat{h}^\star, h^\star) + \mathcal{L}(h) - \mathcal{L}(\widehat{h}^\star)\big) ,$$

where we recall that $\gamma(h, h') = \mathbb{E}[(h(x) - h'(x))^2]$. Setting $\theta = \frac{\beta}{k^2}$ in the above makes the bias smaller than the desired regret bound $\beta$, and yields a variance bound of the form

$$\big(\gamma(\widehat{h}^\star, h^\star) + \mathcal{L}(h) - \mathcal{L}(\widehat{h}^\star)\big) \log \frac{k}{\beta} \quad (4)$$

and a range bound of the form $\log \frac{k}{\beta}$.

In broad strokes, our goal in the ERM analysis is to make sure that with high probability we are able to separate any $h$ from $\widehat{h}^\star$, whenever $h \notin \mathcal{H}_\beta$. This separation effort is made easy as the loss difference $\mathcal{L}(h) - \mathcal{L}(\widehat{h}^\star)$ increases. Yet, at the same time, the bigger $\mathcal{L}(h) - \mathcal{L}(\widehat{h}^\star)$ the higher the variance (4). Solving this trade-off within standard concentration inequalities, and taking a union bound over $h \in \mathcal{H}_\beta$ gives the claimed sample complexity bound. $\square$

Theorem 4.1 improves in a number of ways over the existing literature. For instance, comparing to (Busa-Fekete et al., 2023), we have sharper sample complexity guarantees: our bound reads $n = \widetilde{O}\big(k(\gamma(\widehat{h}^\star, h^\star) + \beta)/\beta^2\big)$, which becomes $n = \widetilde{O}(k/\beta)$ in the realizable case, while the regret guarantees in (Busa-Fekete et al., 2023) are only of the form $n = \widetilde{O}(k^2/\beta^2)$ (that is, slow rates only and, in addition, a quadratic dependence on $k$, instead of linear). On the other hand, the results in (Busa-Fekete et al., 2023) apply to all bounded losses, not just to square loss.

The more recent paper (Li et al., 2024) also covers the square loss case, but the results contained there are widely sub-optimal when it comes to the dependence on $k$. For instance, the authors show fast rates for a debiased square loss algorithm, but the sample complexity therein reads (in our notation) $n = \widetilde{O}(k^3/\beta)$. Besides, the authors consider only a more restricted notion of realizability where $\mathcal{L}(h^\star) = 0$ and the functions in the hypothesis space $\mathcal{H}$ have binary output, thereby ruling out any potential noise in the labels. Needless to say, the assumption about deterministic labels is particularly problematic when we want to associate any notion of privacy with the label aggregation mechanism.

Finally, compared to the optimal bounds in Theorem 3.1, those in Theorem 4.1 are looser only by log factors, but they clearly apply to wider settings.

## 5. Stochastic Gradient Descent

In this section, we show that a similar variance reduction technique as in Section 4 leads to improved results for Stochastic Gradient Descent methods applied to LLP tasks.

For the sake of this section, the loss is still the square loss $\ell(h(x), y) = (y - h(x))^2$, but the hypothesis space is that of norm-bounded $d$-dimensional linear predictors,

$$\mathcal{H} = \{h_w : x \to w \cdot x \mid w \in \mathbb{R}^d, \|w\| \le \rho_w\},$$

with known $\rho_w > 0$. We can think of the labels $y \in \mathcal{Y}$ as binary, as before, but there is no real need for this restriction. So, in this section, we look at the problem more as a regression problem, and assume the labels are real-valued and bounded: $\mathcal{Y} = \{y \in \mathbb{R} : |y| \le \rho_y\}$, for some known $\rho_y > 0$. The label aggregation at each bag $\mathcal{B}_j$ is still done via averaging: $\alpha_j = \frac{1}{k}\sum_{i=1}^k y_{j,i}$. The population loss can now be conveniently expressed as a function of $w$ directly:

$$\mathcal{L}(w) = \mathbb{E}_{(x,y)\sim\mathcal{D}}[\ell(h_w(x), y)] = \mathbb{E}_{(x,y)\sim\mathcal{D}}[(y - w \cdot x)^2]$$

Further, as is customary in SGD analyses, we assume that $\mathcal{D}$ is such that $\|x\| \le \rho_x$ almost surely, and that the constant $\rho_x > 0$ is given to us. For simplicity we shall also assume that the following quantities are known (exactly):

$$\mu_x = \mathbb{E}[x], \quad \mu_y = \mathbb{E}[y].$$

---

**Algorithm 2** SGD-based algorithm

**Input:** stepsize $\eta$, clipping threshold $\gamma > 0$
Initialize at $w_1 = 0$
**for** bags $j = 1, 2, \ldots, m$ **do**
  Compute centroid $\bar{x}_j = \frac{1}{k}\sum_{i=1}^k x_{j,i}$ ;
  **if** $\|\bar{x}_j - \mu_x\| \le \gamma\rho_x$ **then**
    update

$$w_{j+1} = \Pi[w_j - \eta\nabla\ell(w_j, z_j)]$$

  where $\ell(w, z_j)$ is as in the main text.
  **else**
    skip update and proceed to the next step
  **end if**
**end for**
Output $\bar{w}_m = \frac{1}{m}\sum_{j=1}^m w_j$.

---

These are the marginal expectations under $\mathcal{D}$, that can be approximated from (either individual or bagged) data in a straightforward and sample-efficient manner. We emphasize that the method we are about to present can be immediately modified to have $\mu_x$ and $\mu_y$ estimated from data instead (the modification would be similar in spirit to the one contained in Section B.2). Observe that, due to our assumptions, we have $\|\mu_x\| \le \rho_x$ and $\|\mu_y\| \le \rho_y$.

The algorithm we propose and analyze for this setting in presented in Algorithm 2. The algorithm operates with the bag-level square loss

$$\ell(w, z_j) = k\left(w \cdot (\bar{x}_j - \mu_x) - (\alpha_j - \mu_y)\right)^2 + (w \cdot \mu_x - \mu_y)^2$$

which can be easily verified to coincide with the non-clipped bag level loss (2), upon setting $h(x) = w \cdot x$ and $p = \mathbb{E}[y]$. Essentially, the algorithm constructs a debiased version $\ell(\cdot, z_j)$ of the square loss $(y - w \cdot x)^2$ based on each bag $z_j = (\mathcal{B}_j, \alpha_j)$, and performs SGD updates on a truncated (variance-reduced) version of this loss function, defined as

$$\widetilde{\ell}(w, z_j) = \ell(w, z_j)\{\|\bar{x}_j - \mu_x\| \le \theta\rho_x\},$$

for some $\theta > 0$. The updates are projected back to the feasible set of norm-bounded models $\{w \in \mathbb{R}^d : \|w\| \le \rho_w\}$; here $\Pi$ denotes the Euclidean projection operator onto the latter set.

Our main result in this section is the following population excess risk bound for Algorithm 2:

**Theorem 5.1.** *Let $\mathcal{X}, \mathcal{Y}, \mathcal{D}, \mathcal{H}, \mu_x, \mu_y, \rho_w, \rho_x, \rho_y$ be as described earlier in this section. Let a sample $S$ of size $n = mk$, with $m$ bags of size $k$, be drawn i.i.d. according to $\mathcal{D}$. Set*

$$\theta = \sqrt{\frac{8}{k}\log\left(\frac{9km(\rho_w\rho_x + \rho_y)^2}{\rho_w^2\rho_x^2}\right)}$$

*and*

$$\zeta = (k\theta^2 + 1)\rho_x^2 \, .$$

*Then, for any $L^\star > 0$, the hypothesis $\bar{w}_m$ output by Algorithm 2 with step size*

$$\eta = \min\left\{ \frac{\rho_w}{\sqrt{2\zeta m L^\star}}, \frac{1}{2\zeta} \right\}$$

*and clipping threshold $\theta$ satisfies, for all $w^*$ such that $\|w^*\| \le \rho_w$ and $\mathcal{L}(w^*) \le L^\star$, and any $\beta > 0$*

$$\mathbb{E}[\mathcal{L}(\bar{w}_m)] - \mathcal{L}(w^*) \le \beta \, ,$$

*whenever*

$$m = \widetilde{O}\left( \frac{\rho_x^2 \rho_w^2}{\beta} + \frac{L^\star \rho_x^2 \rho_w^2}{\beta^2} \right) \, .$$

*In the above, the expectation is over the generation of $S$, and the asymptotic notation hides logarithmic factors of the form $\log(km)$ and $\log\left(1 + \frac{\rho_y}{\rho_w \rho_x}\right)$.*

In particular, in the nearly realizable case when $L^\star \approx 0$, the theorem gives a fast $O(1/m)$ rate of convergence, as opposed to the $O(1/\sqrt{m})$ rate obtained in the non-realizable case (where $L^\star \gg 0$).

The sample complexity above reads $n = \widetilde{O}(k/\beta^2)$ in the non-realizable case, and $n = \widetilde{O}(k/\beta)$ in the realizable case. It is instructive to compare to the SGD result from Busa-Fekete et al. (2023) (Thm 6.1 therein), whose sample bound is of the form $n = \widetilde{O}(k^2/\beta^2)$. Thus, in addition to obtaining fast rates, we shave a factor $k$ from the sample complexity. On the other hand, it is fair to say that the result in (Busa-Fekete et al., 2023) applies to more general convex losses.

*Proof sketch.* Our basic approach to Algorithm 2 and establishing Theorem 5.1 is to observe that the algorithm performs online projected gradient updates with respect to the loss sequence $\widetilde{\ell}(\cdot, z_1), \ldots, \widetilde{\ell}(\cdot, z_m)$, and to appeal to an online-to-batch argument for obtaining a population-level excess risk bound for this algorithm. Crucially, our analysis leverages the fact that the non-truncated bag-level loss $\ell(w, z_j)$ is an unbiased estimator of the population square loss at the individual example level (recall Equation (3)), as well as the *smoothness* of the (truncated) squared loss functions to obtain fast rates in (nearly) realizable scenarios. □

# 6. Experiments

In this section we compare our proposed LLP loss to baseline losses on several datasets and models following closely the setup of Busa-Fekete et al. (2023).

## 6.1. Experimental Setup

We prepare each training dataset by shuffling the data, partitioning it into consecutive bags of size $k$, and replacing the labels within each bag by their average. Then we train models on the LLP data as follows: On each training epoch (a single pass through the processed training data), we shuffle the order of the bags and group them into batches containing a fixed number $N$ of examples. The examples within each bag are the same on each epoch, only the order of the bags is permuted. On all datasets we choose a batch size of $N = 1024$ and use bag sizes $k = 2^i$ for $i = 0, \ldots, 9$. For each batch, we compute the gradient of the aggregate loss function and use Adam (Kingma & Ba, 2015) to update the model parameters. After completing $E$ training epochs, we evaluate the model's test accuracy at the level of individual examples $(x, y)$.

We repeat the above training procedure 10 times for each loss function, dataset, bag size, and Adam learning rate in the set $\{10^{-6}, 5 \cdot 10^{-6}, 10^{-5}, 10^{-4}, 5 \cdot 10^{-4}, 10^{-3}, 5 \cdot 10^{-3}, 10^{-2}\}$. Each repetition uses a different partition of the training data into bags and different random model initialization. For each bag size and aggregate loss, we report the average test accuracy at the end of $E$ training epochs (that is, cycles through the data) achieved by the best performing learning rate.

## 6.2. Datasets and Models

We conduct our experiments on the following datasets and models. We use versions of MNIST (LeCun et al., 2010) and CIFAR-10 (Krizhevsky, 2009) with binary labels, together with the Higgs (Baldi et al., 2014) and UCI Adult (Kohavi & Becker, 1996) datasets, which are already binary tasks. Following Busa-Fekete et al. (2023), we binarize the labels of MNIST based on whether they are even or odd and, for CIFAR-10, we replace the original labels by whether the image depicts an animal (bird, cat, deer, dog, frog, horse) or machine (airplane, automobile, ship, truck). On MNIST and CIFAR-10 we train CNN models, while on Higgs and UCI Adult we train fully connected networks. Full details of the datasets and models are given in Appendix D.

## 6.3. Aggregate Losses

We train models using five aggregate losses. OURS refers to the unclipped[3] version of our proposed loss, given in (2). LI ET AL. is the aggregate loss proposed by Li et al. (2024), which is also an unbiased estimate of the squared

---

[3] Our choice of using the unclipped loss rather than the clipped one was mainly practical. On one hand, the clipped loss introduces one more parameter to tune (the clipping constant $\theta$). On the other, we observed in preliminary experiments that our method works well even without clipping.

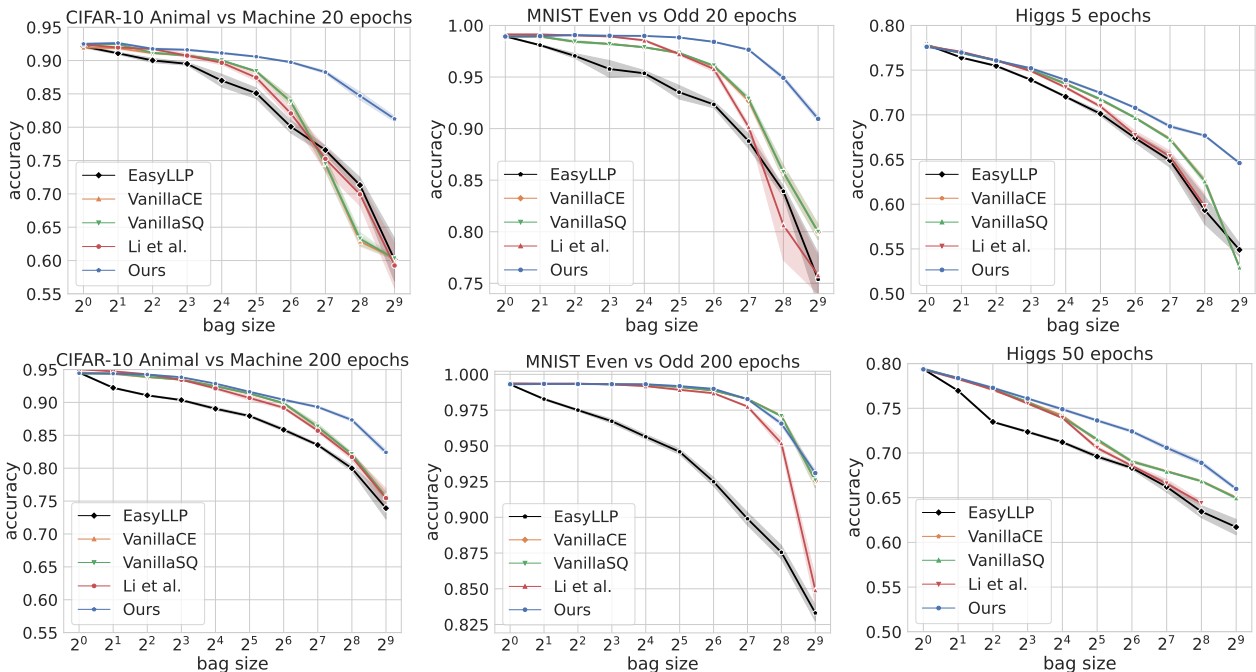

*Figure 1.* Plots showing the final test accuracy of models trained on selected datasets as a function of the LLP bag size and number of training epochs. Error bars show one standard error in the average over 10 repetitions of each run. Each data point represents the accuracy achieved by a learning rate tuned for that bag size and loss. Some of the curves are not visible since they are overlapping. This is the case, in particular, for VANILLACE, which is often overalapped with VANILLASQ.

loss at individual examples, but suffers from higher variance. VANILLASQ and VANILLACE refer to the proportion matching losses where we compare the model's average prediction to the label proportion using squared loss and binary cross-entropy, respectively. Finally, EASYLLP refers to the unbiased loss estimate of Busa-Fekete et al. (2023) applied with the binary cross-entropy loss. Full details of the aggregate losses are given in Appendix D, including the way we implemented the estimation of the unknown quantities $\mathbb{E}[h(x)]$ and $p$ for OURS.

### 6.4. Results

Figure 1 shows the final test accuracy of each aggregate loss at various bag sizes and number of training epochs (some of the curves are not visible since they are overlapping to one another). The aggregate losses OURS, LI ET AL., and EASYLLP are all unbiased estimates of the population loss at the level of individual examples. However, as we will see in Section 6.5, the variance of OURS grows significantly slower with the bag size $k$ than that of LI ET AL. and EASYLLP. In comparison, the VANILLA baselines are not unbiased. At every bag size and training duration, our proposed method achieves the highest accuracy. In comparison, even though the other methods perform reasonably well at low bag sizes, once $k$ is sufficiently large, the high variance of these method eventually causes them to perform

far worse worse.[4] We also note that OURS tends to widely outperform the other baselines especially when the number of training epochs is lower, suggesting that these losses lead to more computationally efficient training.

As for comparison to the method proposed by Dulac-Arnold et al. (2019), we note that their method has already been shown to underperform in comparable experimental setups. For instance, on CIFAR-10, Busa-Fekete et al. (2023) report a test set accuracy of around 0.65 with bag size $k = 2^6$ (see Figure 4 therein), while all the methods we tested are above 0.8 on a similar CNN architecture. On Higgs, with the very same NN architecture (the one originating from (Baldi et al., 2014)), all the methods we tested have test set accuracy which is above by at least 5% across all large enough bag sizes. E.g., for $k = 2^6$, Busa-Fekete et al. (2023) report a test accuracy below 0.6 (see Figure 6 therein), while all our methods are above 0.65.

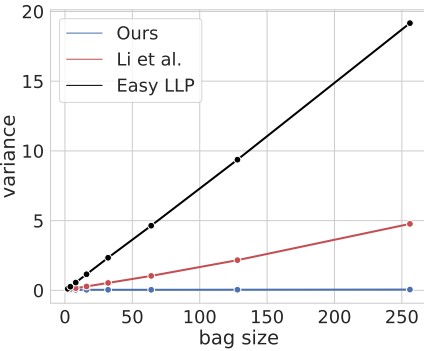

*Figure 2.* Plot showing how the variance of the loss estimate of OURS, LI ET AL and EASYLLP, grows with the bag size in the simple setting described in Section 6.5. Both LI ET AL. and EASYLLP have variacne that grows linearly with the bag size, while OURS has constant variance, close to zero.

## 6.5. Variance comparison

In this section we compare the variance of the loss estimates produced by our method, LI ET AL., and EASYLLP for a simple synthetic data distribution and model $\hat{h}$ defined below. Recall that all three of these methods produce unbiased estimates of the squared loss on individual examples, so the variance characterizes how reliable those estimates are. We showed that the variance of the loss estimate computed from a single bag using our method does not grow with $k$, while the variance of all prior methods does.

We consider the simple setting where $x$ is uniform in $[0, 1]$, and the distribution of $y$ is $\Pr(y = 1 \mid x) := h^*(x) = x^2$. We estimate the loss of the model $\hat{h}(x) = x$. Each method is used to estimate $\mathbb{E}[(y - \hat{h}(x))^2]$ based on LLP data. We compute the label marginal $p$ exactly, and estimate $\mathbb{E}[\hat{h}(x)]$ in the same way as our learning experiments. That is, we break that data into batches containing 1024 examples (regardless of bag size) and for each bag within the batch, we estimate $\mathbb{E}[\hat{h}(x)]$ as the sample average of $\hat{h}(x)$ over the batch excluding that bag.

In Figure 2 we plot the empirical variance of the loss estimates computed from $n = 2^{20}$ examples worth of bags for bags of size $k = 2, 4, 8, 16, 32, 64, 128, 256$. We see that both LI ET AL. and EASYLLP have empirical variance that grows linearly with the bag size, while OURS is essentially constantly equal to 0.034. We remark that the variance of OURS does grow slightly to 0.055 at bag size $k = 256$, but this is because, as the bag size grows, there are fewer remaining examples within the batch to estimate $\mathbb{E}[\hat{h}(x)]$.

---

[4] In this respect, we need to point out that our original implementation of the method from (Li et al., 2024) did contain a measurement bug that made it perform better than reported here. This bug was contained in the original submission of this paper, and has now been fixed.

## 7. Conclusions, Limitations, and Ongoing Research

We have studied LLP in the relevant case where bags are drawn i.i.d. according to a fixed but unknown distribution over $\mathcal{X} \times \mathcal{Y}$. Via suitable variance reduction techniques, we have proven tight bounds on the sample complexity in realizable and non-realizable settings, with substantial improvements over prior art in comparable statistical learning settings.

We have empirically contrasted our variance reduction methods to available LLP baselines on a variety of datasets and underlying model architectures, showing superior test set performance at the individual example level. The performance gap is especially remarkable for big bag sizes, the regime that matters the most for LLP applications.

As for current limitations, this paper could be extended along a number of directions.

- We would like to achieve regret guarantees also for practically relevant loss functions beyond square loss, like the log loss for ERM and more general convex losses than square loss for SGD. The extension to log loss, for instance, is nontrivial, as the log loss is itself an *unbounded* loss function. This is a loss function commonly adopted in DNN training.

- As currently presented, our analysis only applies to binary classification. A suitable extension to multiclass classification would be important in practice.

- Another practically important extension is the case when the examples within the bags or the bags themselves come from distributions that drifts over time. This is another practically relevant scenario where a low variance estimator is expected to have an edge, thanks to its higher adaptivity.

- There are also technical limitations related to some of the formal statements we provided. For instance, Theorem 3.1 applies only when $k$ is big enough ($k = \Omega(1/\beta)$), not for smaller $k$.

## Impact Statement

This paper presents work whose goal is to advance the field of Machine Learning. There are many potential societal consequences of our work, none which we feel must be specifically highlighted here.

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

## A. Proofs for Section 3

This appendix contains the proofs for the theorems contained in the warmup section.

### A.1. Proof of Theorem 3.1

In order to quantify the probability of error of Algorithm 1, suppose $|\Delta| \geq \sqrt{\frac{\beta}{2k}}$, with $h^\star = h_1$. Note that we can rewrite

$$A_1 = \sum_{j=1}^{m} \sum_{i=1}^{k} (y_{j,i} - h_1(x_{j,i})) \qquad A_2 = \sum_{j=1}^{m} \sum_{i=1}^{k} (y_{j,i} - h_2(x_{j,i})) \,.$$

Also, $\mathbb{E}_{(x,y)}[y - h_1(x)] = 0$ while $\mathbb{E}_{(x,y)}[y - h_2(x)] = \Delta$. Set $C_\delta(m, k) = \sqrt{\frac{mk}{2} \log \frac{4}{\delta}}$. Then, from a standard concentration inequality (Hoeffding) we have, with probability at least $1 - \delta$ over the random draw of $S$,

$$|A_1(S)| \leq C_\delta(m, k) \qquad \text{and} \qquad |A_2(S) - mk\Delta| \leq C_\delta(m, k)$$

so that, with the same probability,

$$|A_1(S)| \leq C_\delta(m, k) \qquad \text{and} \qquad |A_2(S)| \geq mk|\Delta| - C_\delta(m, k) > C_\delta(m, k) \,,$$

the last inequality holding provided $mk > \frac{2}{\Delta^2} \log \frac{4}{\delta}$.

A symmetric argument holds if $h^\star = h_2$. We conclude that if $|\Delta| \geq \sqrt{\frac{\beta}{2k}}$ the total sample size $n = mk$ needed to insure $\mathbb{P}(\widehat{h} \neq h^\star) \leq \delta$ is

$$n = O\left( \frac{1}{\Delta^2} \log \frac{1}{\delta} \right) \,.$$

On the other hand, in the sequel we also show that when $|\Delta| < \sqrt{\frac{\beta}{2k}}$, the sample size $n = mk$ also satisfies

$$n = O\left( \frac{k}{\beta} \log \frac{1}{\delta} \right) \,.$$

Since $\frac{1}{\Delta^2} < \frac{2k}{\beta}$ (that is, the first bound is better than the second) if and only if $|\Delta| \geq \sqrt{\frac{\beta}{2k}}$, this will show that the overall sample size indeed satisfies

$$n = O\left( \min\left\{ \frac{1}{\Delta^2}, \frac{k}{\beta} \right\} \log \frac{1}{\delta} \right) \,.$$

We now continue by analyzing the probability of error when Algorithm 1 relies on the sign of $Q(S)$, under the condition $|\Delta| < \sqrt{\frac{\beta}{2k}}$.

Denote for brevity by $\mathbb{P}'(\cdot)$ the conditional probability $\mathbb{P}(\cdot \mid \mathcal{B}_1, \ldots, \mathcal{B}_m)$, that is, the probability on the labels conditioned on the values of the $x_{j,i}$ on all bags $\mathcal{B}_j = \{x_{j,1}, \ldots, x_{j,k}\}$ in $S$.

Suppose $\Delta^2 \leq \beta/2$ (which is implied by the condition $|\Delta| < \sqrt{\frac{\beta}{2k}}$) and $k \geq \frac{2}{\Phi^2(-1)\beta}$, where $\Phi(t)$ is the cumulative distribution function of the standard Gaussian distribution.

We say that $\mathcal{B}_j$ is *discriminative* for $\{h_1, h_2\}$ at threshold $\theta$ if

$$\mathbb{E}_{j,1} \geq \mathbb{E}_{j,2} + \theta \qquad \text{or} \qquad \mathbb{E}_{j,2} \geq \mathbb{E}_{j,1} + \theta \,,$$

for a suitable $\theta = \theta(\beta, k, \delta) > 0$ that we set later on. Let us rewrite $Q$ so as to single out discriminative and nondiscriminative bags:

$$Q = \sum_{j=1}^{m} \{\mathbb{E}_{j,1} \leq \mathbb{E}_{j,2} - \theta\}(k\alpha_j - \mu_j) + \sum_{j=1}^{m} \{\mathbb{E}_{j,2} - \theta < \mathbb{E}_{j,1} \leq \mathbb{E}_{j,2}\}(k\alpha_j - \mu_j)$$

$$+ \sum_{j=1}^{m} \{\mathbb{E}_{j,1} > \mathbb{E}_{j,2} + \theta\}(\mu_j - k\alpha_j) + \sum_{j=1}^{m} \{\mathbb{E}_{j,2} + \theta \geq \mathbb{E}_{j,1} > \mathbb{E}_{j,2}\}(\mu_j - k\alpha_j) \,.$$

Assume $h^\star = h_1$. We want then to bound the probability that $Q \geq 0$ (that is, the probability that we make the wrong decision).

Now,

$$
\begin{aligned}
\{\mathbb{E}_{j,1} \leq \mathbb{E}_{j,2} - \theta\}(k\alpha_j - \mu_j) &\leq \{\mathbb{E}_{j,1} \leq \mathbb{E}_{j,2} - \theta\}(k\alpha_j - \mathbb{E}_{j,1} - \theta/2) \\
\{\mathbb{E}_{j,2} - \theta < \mathbb{E}_{j,1} \leq \mathbb{E}_{j,2}\}(k\alpha_j - \mu_j) &\leq \{\mathbb{E}_{j,2} - \theta < \mathbb{E}_{j,1} \leq \mathbb{E}_{j,2}\}(k\alpha_j - \mathbb{E}_{j,1}) \\
\{\mathbb{E}_{j,1} > \mathbb{E}_{j,2} + \theta\}(\mu_j - k\alpha_j) &\leq \{\mathbb{E}_{j,1} > \mathbb{E}_{j,2} + \theta\}(\mathbb{E}_{j,1} - \theta/2 - k\alpha_j) \\
\{\mathbb{E}_{j,2} + \theta \geq \mathbb{E}_{j,1} > \mathbb{E}_{j,2}\}(\mu_j - k\alpha_j) &\leq \{\mathbb{E}_{j,2} + \theta \geq \mathbb{E}_{j,1} > \mathbb{E}_{j,2}\}(\mathbb{E}_{j,1} - k\alpha_j),
\end{aligned}
$$

so that

$$
\mathbb{P}'(Q \geq 0) \leq \mathbb{P}'(Q_1 \geq 0),
$$

where

$$
\begin{aligned}
Q_1 = &\sum_{j=1}^m \{\mathbb{E}_{j,1} \leq \mathbb{E}_{j,2} - \theta\}(\Sigma_{j,1} - \theta/2) + \sum_{j=1}^m \{\mathbb{E}_{j,2} - \theta < \mathbb{E}_{j,1} \leq \mathbb{E}_{j,2}\}\Sigma_{j,1} \\
&+ \sum_{j=1}^m \{\mathbb{E}_{j,1} > \mathbb{E}_{j,2} + \theta\}(-\Sigma_{j,1} - \theta/2) + \sum_{j=1}^m \{\mathbb{E}_{j,2} + \theta \geq \mathbb{E}_{j,1} > \mathbb{E}_{j,2}\}(-\Sigma_{j,1})
\end{aligned}
$$

being $\Sigma_j = k\alpha_j - \mathbb{E}_{j,1}$ a (conditionally) zero-mean random variable. Introduce the short-hands

$$
m_1 = \sum_{j=1}^m \{\mathbb{E}_{j,1} \leq \mathbb{E}_{j,2} - \theta\} + \sum_{j=1}^m \{\mathbb{E}_{j,1} > \mathbb{E}_{j,2} + \theta\}
$$

$$
m_2 = \sum_{j=1}^m \{\mathbb{E}_{j,2} - \theta < \mathbb{E}_{j,1} \leq \mathbb{E}_{j,2}\} + \sum_{j=1}^m \{\mathbb{E}_{j,2} + \theta \geq \mathbb{E}_{j,1} > \mathbb{E}_{j,2}\},
$$

and note that the above quantities are fully determined by $\mathcal{B}_1, \ldots, \mathcal{B}_m$. Also, $m_1 + m_2 = m$ for all $\theta \geq 0$.

Consider the four sums defining $Q_1$ as aggregated in pairs according to the definition of $m_1$ and $m_2$ above. Observe that the random variables $y_{j,1}, y_{j,2}, \ldots, y_{j,k}$, for all $j$ such that $\mathbb{E}_{j,1} \leq \mathbb{E}_{j,2} - \theta$ or such that $\mathbb{E}_{j,1} > \mathbb{E}_{j,2} + \theta$ are $m_1 k$-many conditionally independent Bernoulli random variables, each with its own bias. Hence we can apply standard concentration bounds (Hoeffding's inequality) to conclude that

$$
\begin{aligned}
&\mathbb{P}'\left(\sum_{j=1}^m \{\mathbb{E}_{j,1} \leq \mathbb{E}_{j,2} - \theta\}(\Sigma_{j,1} - \theta/2) + \sum_{j=1}^m \{\mathbb{E}_{j,1} > \mathbb{E}_{j,2} + \theta\}(-\Sigma_{j,1} - \theta/2) > t_1\right) \\
&= \mathbb{P}'\left(\sum_{j=1}^m \{\mathbb{E}_{j,1} \leq \mathbb{E}_{j,2} - \theta\}\Sigma_{j,1} - \sum_{j=1}^m \{\mathbb{E}_{j,1} > \mathbb{E}_{j,2} + \theta\}\Sigma_{j,1} > m_1\theta/2 + t_1\right) \\
&\leq \exp\left(-\frac{2(m_1\theta/2 + t_1)^2}{m_1 k}\right)
\end{aligned}
$$

for all $t_1 > -m_1\theta/2$. Similarly,

$$
\mathbb{P}'\left(\sum_{j=1}^m \{\mathbb{E}_{j,2} - \theta < \mathbb{E}_{j,1} \leq \mathbb{E}_{j,2}\}\Sigma_{j,1} - \sum_{j=1}^m \{\mathbb{E}_{j,2} + \theta \geq \mathbb{E}_{j,1} > \mathbb{E}_{j,2}\}\Sigma_{j,1} > t_2\right) \leq \exp\left(-\frac{2t_2^2}{m_2 k}\right)
$$

for all $t_2 > 0$.

Therefore, in order to bound $\mathbb{P}'(Q_1 \geq 0)$, it suffices to select $t_1$ and $t_2$ in the above in such a way that $t_1 + t_2 = 0$, with the constraints $t_1 > -m_1\theta/2$, and $t_2 > 0$, allowing us to conclude that

$$
\mathbb{P}'(Q \geq 0) \leq \exp\left(-\frac{2(m_1\theta/2 + t_1)^2}{m_1 k}\right) + \exp\left(-\frac{2t_2^2}{m_2 k}\right).
$$

The above constraints are easily satisfied if $m_1$ and $m_2$ are constant multiples of one another. For instance, suppose there exist positive constants $c_1$ and $c_2$ such that $m_i = c_i m$, and $c_1 + c_2 = 1$. Then we may set

$$t_1 = -\frac{c_1 m\theta}{4}, \quad t_2 = \frac{c_1 m\theta}{4}$$

obtaining

$$\mathbb{P}'(Q \geq 0) \leq \exp\left(-\frac{c_1\theta^2 m}{8k}\right) + \exp\left(-\frac{c_1^2\theta^2 m}{8c_2 k}\right) \tag{5}$$

which is of the form $\exp\left(-\frac{\theta^2 m}{k}\right)$, provided $c_1$ and $c_2$ are constants independent of $m$ and $\theta$.

We expect to be able to set $\theta$ to be of the form $\theta = \beta\sqrt{k}$ via an anti-concentration argument.

Now we lower bound via anti-concentration the probability of drawing a bag $\mathcal{B}_j$ that is discriminative for $\{h_1, h_2\}$. Recall that $\Delta = \mathbb{E}[h_1(x)] - \mathbb{E}[h_2(x)]$. We can write

$$\{\mathbb{E}_{j,1} \geq \mathbb{E}_{j,2} + \theta\} + \{\mathbb{E}_{j,1} < \mathbb{E}_{j,2} - \theta\} = \left\{\sum_{x\in B_j}(h_1(x) - h_2(x)) \geq \theta\right\} + \left\{\sum_{x\in B_j}(h_2(x) - h_1(x)) \geq \theta\right\}.$$

Hence, we are left to *lower* bound the expectation of the indicators above. Now, the variables

$$Z_{j,i} = h_1(x_{j,i}) - h_2(x_{j,i}), \quad i = 1, \ldots, k,$$

are i.i.d. random variables with range $[-1, 1]$ and expectation $\Delta \in [-1, 1]$. Thus we are compelled to leverage anti-concentration inequalities for sum of $[-1, 1]$-valued i.i.d. random variables.

Since we are working under the assumption $\Delta^2 \leq \beta/2$ and $k \geq \frac{2}{\Phi^2(-1)\beta}$, an easy route is to go through Berry-Esseen's finite sample version of the Central Limit Theorem, and then rely on the anti-concentration of standard normal variables. One version of Berry-Esseen's theorem claims that for $\Sigma = \sum_{i=1}^k X_i$, where $X_1, \ldots, X_k$ are i.i.d. with zero mean, variance $\sigma^2 = \mathbb{E}[X^2]$, and finite third moment $\rho = \mathbb{E}[|X|^3]$ we have

$$\sup_t |\mathbb{P}(\Sigma \leq t\sigma\sqrt{k}) - \Phi(t)| \leq \frac{\rho}{2\sigma^3\sqrt{k}},$$

where $\Phi(t)$ is the cumulative distribution function of the standard Gaussian. Note that by convexity $\rho \geq \sigma^3$. Moreover, $\rho \leq \sigma^2$ since in our case $|X| \leq 1$.

We have

$$\mathbb{E}\left[\{\mathbb{E}_{j,1} \geq \mathbb{E}_{j,2} + \theta\}\right] + \mathbb{E}\left[\{\mathbb{E}_{j,1} < \mathbb{E}_{j,2} - \theta\}\right] = \mathbb{P}\left(\sum_{i=1}^k Z_{j,i} \geq \theta\right) + \mathbb{P}\left(\sum_{i=1}^k Z_{j,i} < -\theta\right)$$

$$= \mathbb{P}\left(\sum_{i=1}^k (Z_{j,i} - \Delta) \geq \theta - k\Delta\right) + \mathbb{P}\left(\sum_{i=1}^k (Z_{j,i} - \Delta) < -\theta - k\Delta\right)$$

$$\geq \Phi\left(-\frac{\theta - k\Delta}{\sigma\sqrt{k}}\right) + \Phi\left(-\frac{\theta + k\Delta}{\sigma\sqrt{k}}\right) - \frac{2\rho}{2\sigma^3\sqrt{k}}$$

$$\geq \Phi\left(-\frac{\theta - k\Delta}{\sigma\sqrt{k}}\right) + \Phi\left(-\frac{\theta + k\Delta}{\sigma\sqrt{k}}\right) - \frac{1}{\sqrt{\sigma^2 k}}$$

$$\text{(since } \rho \leq \sigma^2\text{)},$$

$$\geq \Phi\left(-\frac{\theta - k\Delta}{\sigma\sqrt{k}}\right) + \Phi\left(-\frac{\theta + k\Delta}{\sigma\sqrt{k}}\right) - \sqrt{\frac{2}{\beta k}} \tag{6}$$

$$\text{(since } \sigma^2 = \beta - \Delta^2 \geq \beta/2, \text{ by the assumption } \Delta^2 \leq \beta/2\text{)},$$

where
$$\sigma^2 = \mathbb{E}[(Z_{j,i} - \Delta)^2], \qquad \rho = \mathbb{E}[|Z_{j,i} - \Delta|^3] \,.$$

We set $\theta = k|\Delta| + \sigma\sqrt{k}$. With this setting of $\theta$, if $\Delta \geq 0$ then $\Phi\left(-\frac{\theta - k\Delta}{\sigma\sqrt{k}}\right) = \Phi(-1)$. On the other hand, if $\Delta < 0$, then $\Phi\left(-\frac{\theta + k\Delta}{\sigma\sqrt{k}}\right) = \Phi(-1)$. In both cases

$$(6) \geq \Phi(-1) - \sqrt{\frac{2}{\beta\,k}} \geq \Phi(-1)/2 \,,$$

the last inequality using $k \geq \frac{2}{\Phi^2(-1)\beta}$. This yields

$$\mathbb{E}\left[\{\mathbb{E}_{j,1} \geq \mathbb{E}_{j,2} + \theta\}\right] + \mathbb{E}\left[\{\mathbb{E}_{j,1} < \mathbb{E}_{j,2} - \theta\}\right] \geq \Phi(-1)/2 \,.$$

Thus, on a sample $S$ made up of $m$ i.i.d. bags, on average, a fraction of at least $m\Phi(-1)/2$ bags will be discriminative at threshold $\theta = k|\Delta| + \sigma\sqrt{k}$, that is,

$$\mathbb{E}_{\mathcal{B}_1,\ldots,\mathcal{B}_m \sim \mathcal{D}_{\mathcal{X}}^{mk}}[m_1] \geq m\Phi(-1)/2 \,.$$

Since the $m$ bags are i.i.d., the random variables $\{\mathbb{E}_{j,1} \geq \mathbb{E}_{j,2} + \theta\} + \{\mathbb{E}_{j,1} < \mathbb{E}_{j,2} - \theta\}$ are i.i.d. Bernoulli variables (w.r.t. the random draws of $\mathcal{B}_1, \ldots, \mathcal{B}_m$). From standard multiplicative Chernoff bounds, this implies that

$$\mathbb{P}_{\mathcal{B}_1,\ldots,\mathcal{B}_m \sim \mathcal{D}_{\mathcal{X}}^{mk}}\left(m_1 \leq \frac{m}{4}\Phi(-1)\right) \leq \exp\left(-\frac{m\,\Phi(-1)}{16}\right) \,.$$

In order to conclude, we get back to (5). Combining with the above yields

$$\mathbb{E}[\{Q \geq 0\}] \leq \mathbb{E}\left[\{Q \geq 0, m_1 > \frac{m}{4}\Phi(-1)\}\right] + \mathbb{E}\left[\{m_1 \leq \frac{m}{4}\Phi(-1)\}\right]$$
$$\leq e^{-\Theta(\theta^2 m/k)} + e^{-\Theta(m)} \,.$$

We want to make the above smaller than $\delta$. It suffices to pick

$$m = \left(1 + \frac{k}{\theta^2}\right)\log\frac{2}{\delta} \,.$$

Now, observe how $\theta$ depends on the separation parameter $\beta$. Note that $\beta = \mathbb{E}[Z_{j,i}^2]$, and recall that $\sigma^2 = \beta - \Delta^2$. We have

$$\frac{\theta^2}{k} = k\Delta^2 + 2|\Delta|\sigma\sqrt{k} + \sigma^2$$
$$= (k-1)\Delta^2 + 2|\Delta|\sigma\sqrt{k} + \beta$$
$$\geq (k-1)\Delta^2 + \beta$$
$$\geq \beta \,,$$

independent of the bias $\Delta$ (and in fact, we have equality to $\beta$ if $\Delta = 0$). Thus the sample complexity $n = mk$ becomes

$$n = O\left(\frac{k}{(k-1)\Delta^2 + \beta}\log\frac{1}{\delta}\right) = O\left(\min\left\{\frac{1}{\Delta^2}, \frac{k}{\beta}\right\}\log\frac{1}{\delta}\right) \,.$$

The case $Q < 0$ with $h^\star = h_2$ is treated similarly, since it is completely symmetric. This proves the first part of Theorem 3.1

In order to prove the second half, consider the following simple problem. Let the input space $\mathcal{X}$ be binary, $\mathcal{X} = \{0, 1\}$, and the hypothesis space $\mathcal{H}$ be $\mathcal{H} = \{h_1, h_2\}$, where $h_1, h_2 : \mathcal{X} \to [0, 1]$. Let the marginal distribution of $x$ be Bernoulli(1/2). Moreover, given parameter $\beta \in [0, 1]$, the conditional distribution $h^\star(x) = \eta(x) = \mathbb{P}(y = 1 \mid x)$ is

$$\begin{cases} \mathbb{P}(y = 1 \mid x = 1) = 1/2 + \sqrt{\beta}/2 & \mathbb{P}(y = 1 \mid x = 0) = 1/2 - \sqrt{\beta}/2 & \text{if } h^\star = h_1 \\ \mathbb{P}(y = 1 \mid x = 1) = 1/2 - \sqrt{\beta}/2 & \mathbb{P}(y = 1 \mid x = 0) = 1/2 + \sqrt{\beta}/2 & \text{if } h^\star = h_2 \,. \end{cases}$$

Notice that this implies

$$\mathbb{P}(y = 1) = \frac{1}{2} \mathbb{P}(y = 1 \,|\, x = 1) + \frac{1}{2} \mathbb{P}(y = 1 \,|\, x = 0) = \frac{1}{2}$$

for both $h^\star = h_1$ and $h^\star = h_2$.

Moreover, for square loss,

$$|\mathcal{L}(h_1) - \mathcal{L}(h_2)| = \mathbb{E}[(h_1(x) - h_2(x))^2] = \frac{1}{2}(h_1(1) - h_2(1))^2 + \frac{1}{2}(h_1(0) - h_2(0))^2 = \beta \,.$$

Consider an i.i.d. sample of $m$ bags of size $k$:

$$S = \underbrace{((x_{1,1}, x_{1,2}, \ldots, x_{1,k}), \alpha_1)}_{(\mathcal{B}_1, \alpha_1)}, \ldots, \underbrace{((x_{m,1}, x_{m,2}, \ldots, x_{m,k}), \alpha_m)}_{(\mathcal{B}_m, \alpha_m)} \,,$$

and any function $\widehat{h} : \{S\} \to \mathcal{H}$ that maps any such sample to $\mathcal{H}$. We consider the amount of information needed to perform the inference of separating $h_1$ from $h_2$ out of $S$ through the standard tools of (Shannon) information theory. Let then equip $h^\star$ with a prior distribution

$$h^\star = \begin{cases} h_1 & \text{w.p. } 1/2 \\ h_2 & \text{w.p. } 1/2 \end{cases}$$

and view $h^\star$ itself as a random variable. Consider the mutual information $I(h^\star, \widehat{h})$. By the so-called data-processing inequality and the chain rule of mutual information we have

$$I(h^\star, \widehat{h}) \leq I(h^\star, S) = \sum_{i=1}^{m} I(h^\star; (\mathcal{B}_j, \alpha_i) \,|\, (\mathcal{B}_1, \alpha_1), \ldots, (B_{i-1}, \alpha_{i-1})) \,.$$

Now, since $(\mathcal{B}_j, \alpha_i)$ is conditionally independent of $(\mathcal{B}_1, \alpha_1), \ldots, (B_{j-1}, \alpha_{j-1})$ given[5] $h^\star$, we also have, for all $j$,

$$I(h^\star; (\mathcal{B}_j, \alpha_i) \,|\, (\mathcal{B}_1, \alpha_1), \ldots, (B_{j-1}, \alpha_{j-1})) \leq I(h^\star; (\mathcal{B}_j, \alpha_i))$$

so that from the above we can write

$$I(h^\star, \widehat{h}) \leq m \, I(h^\star; (\mathcal{B}_1, \alpha_1)) \,.$$

Moreover, by Fano's inequality,

$$I(h^\star, \widehat{h}) \geq 1 - \mathcal{E}(\mathbb{P}_e) \,,$$

where $\mathcal{E}(\cdot)$ is the binary entropy function

$$\mathcal{E}(x) = -x \log_2 x - (1 - x) \log_2(1 - x) \,,$$

and $\mathbb{P}_e$ is the "probability of error" $\mathbb{P}_e = \mathbb{P}(\widehat{h} \neq h^\star)$. If we stipulate that we want the target excess risk $\beta$ to be achieved with probability $\geq 1 - \delta$, with $\delta < 1/2$, this implies $\mathbb{P}_e \leq \delta$ and $I(h^\star, \widehat{h}) \geq 1 - \mathcal{E}(\delta)$.

Putting together

$$1 - \mathcal{E}(\delta) \leq m I(h^\star, (\mathcal{B}_1, \alpha_1))$$

which immediately yields the lower bound

$$m \geq \frac{1 - \mathcal{E}(\delta)}{I(h^\star; (\mathcal{B}_1, \alpha_1))} \,. \tag{7}$$

---

[5]This is sometimes phrased by saying that these three variables form a Markov chain

$$(\mathcal{B}_1, \alpha_1), \ldots, (\mathcal{B}_{j-1}, \alpha_{j-1}) \to h^\star \to (\mathcal{B}_j, \alpha_j) \,,$$

see, e.g., Chapter 2 in (Cover & Thomas, 2006).

We are then left with the problem of finding an upper bound on $I(h^\star, (\mathcal{B}_1, \alpha_1))$. Using the chain rule of mutual information, together with $I(X;Y) = H(Y) - H(Y|X)$, where $H$ denotes the Shannon entropy, we can write

$$I(h^\star; (\mathcal{B}_1, \alpha_1)) = \underbrace{I(h^\star; \mathcal{B}_1)}_{=0} + I(h^\star; \alpha_1 \,|\, \mathcal{B}_1)$$

$$= H(\alpha_1 \,|\, \mathcal{B}_1) - H(\alpha_1 \,|\, \mathcal{B}_1, h^\star) \,.$$

Now, since

$$\mathbb{P}(y = 1 \,|\, x) = \mathbb{P}(y = 1\ h^\star = h_1 \,|\, x) + \mathbb{P}(y = 1\ h^\star = h_2 \,|\, x)$$

$$= \frac{1}{2}\, \mathbb{P}(y = 1 \,|\, h^\star = h_1,\, x) + \frac{1}{2}\, \mathbb{P}(y = 1 \,|\, h^\star = h_2,\, x)$$

$$= \frac{1}{2}$$

independent of $x \in \{0,1\}$, the distribution of $\alpha_1$ given $\mathcal{B}_1$ will be the distribution of a (scaled) $Binomial(k, 1/2)$, and its entropy $H(\alpha_1 \,|\, \mathcal{B}_1)$ will be the same as the entropy of a $Binomial(k, 1/2)$.

The following theorem from (Adell et al., 2010) provides useful bounds on it.

**Theorem A.1.** *Let* $X \sim Binomial(k, p)$*, and* $q = 1 - p$*. Then, for* $p \in (0,1)$*,*[6]

$$H(X) = \frac{1}{2} \log_2(2\pi e k p q) + O\left(\frac{1}{k}\right) \,.$$

Thus,

$$H(\alpha_1 \,|\, \mathcal{B}_1) = \frac{1}{2} \log_2(\pi e k / 2) + O\left(\frac{1}{k}\right).$$

Also, the following upper and lower bounds from (Harremoës, 2001) (Theorem 6 and Theorem 7 therein) on the entropy of a Poisson-Binomial distribution[7] $PBin$ in terms of the entropy of related binomial distributions will be useful.

**Theorem A.2.** *Let* $X \sim PBin(p_1, \ldots, p_k)$*, set* $\mu = \sum_{i=1}^{k} p_i$*, and* $\bar{k} = \lfloor \frac{\mu}{p_{\max}} \rfloor$*, where* $p_{\max} = \max_i p_i$*. Let* $\bar{B} \sim Binomial(\bar{k}, \mu/\bar{k})$*, and* $B \sim Binomial(k, \mu/k)$ *Then*

$$\frac{1}{2} \log_2(2\pi e \mu(1 - \mu/k)) + O\left(\frac{1}{k}\right) = H(B) \geq H(X) \geq H(\bar{B}) = \frac{1}{2} \log_2(2\pi e \mu(1 - \mu/\bar{k})) + O\left(\frac{1}{\bar{k}}\right) \,,$$

*the equalities coming from Theorem A.1.*

As for $H(\alpha_1 \,|\, \mathcal{B}_1, h^\star)$, consider the following argument about the conditional distribution of $\alpha_1$ given $\mathcal{B}_1$ and $h^\star$. Let $\mathcal{B}_1$ be a Boolean vector containing $s$ ones and $k - s$ zeroes. The position of these zeros and ones will be immaterial, so for definiteness, let us visualize $\mathcal{B}_1$ as follows:

$$\mathcal{B}_1 = [\underbrace{11\ldots1}_{s}\underbrace{00\ldots0}_{k-s}] \,. \tag{8}$$

Suppose $h^\star = h_1$, and let $Y_1^{(s)}$ and $Y_2^{(s)}$ be the random variables counting the number of $y_i = 1$ in bag $\mathcal{B}_1$ among its first $s$ components, and among its last $k - s$ components, respectively. Since $h^\star = h_1$, we clearly have

$$Y_1^{(s)} \sim Binomial(s, 1/2 + \beta) \qquad \text{and} \qquad Y_2^{(s)} \sim Binomial(k - s, 1/2 - \beta) \,,$$

the two variables being independent. Moreover, $k\alpha = \sum_{i=1}^{k} y_i = Y_1^{(s)} + Y_2^{(s)}$ has distribution

$$PBin(\underbrace{1/2 + \sqrt{\beta}/2, \ldots, 1/2 + \sqrt{\beta}/2}_{s}, \underbrace{1/2 - \sqrt{\beta}/2, \ldots, 1/2 - \sqrt{\beta}/2}_{k-s}) \,.$$

---

[6](Adell et al., 2010) also contains non-asymptotic upper and lower bounds, which are not reported here for brevity.

[7]A Poisson-Binomial random variable with parameters $p_1, \ldots, p_k$ is the sum of $k$ independent Bernoulli random variables, where the $i$-th Bernoulli variable has bias $p_i$.

When $h^\star = h_2$ it is easy to see that we simply have to swap $\sqrt{\beta}/2$ with $-\sqrt{\beta}/2$ in the two binomials, so that $k\alpha \sim PBin(\underbrace{1/2 - \sqrt{\beta}/2, \ldots, 1/2 - \sqrt{\beta}/2}_{s}, \underbrace{1/2 + \sqrt{\beta}/2, \ldots, 1/2 + \sqrt{\beta}/2}_{k-s})$.

Consider the situation when $s = \sum_i x_i$ is such that $s \in A$, where

$$A = \left\{ s \ : \ k/2 - \sqrt{k \log k}/2 \leq s \leq k/2 + \sqrt{k \log k}/2 \right\} , \tag{9}$$

so that when $k \to \infty$ both $s$ and $k - s$ diverge.

By Theorem A.2 (lower bound side), when $h^\star = h_1$ and $s \in A$ we have

$$H(\alpha_1 \mid s, h_1) = H\left( \alpha \,\Big|\, \sum_i x_i = s, h^\star = h_1 \right) \geq \frac{1}{2} \log_2 \left( 2\pi e \mu_1 \left( 1 - \frac{\mu_1}{\bar{k}_1} \right) \right) + O\left( \frac{1}{\bar{k}_1} \right) ,$$

where

$$\bar{k}_1 = \left\lfloor \frac{k + \sqrt{\beta}(2s - k)}{1 + \sqrt{\beta}} \right\rfloor , \qquad \mu_1 = k/2 + \sqrt{\beta}(2s - k)/2 .$$

and

$$H(\alpha_1 \mid s, h_2) = H\left( \alpha \,\Big|\, \sum_i x_i = s, h^\star = h_2 \right) \geq \frac{1}{2} \log_2 \left( 2\pi e \mu_2 \left( 1 - \frac{\mu_2}{\bar{k}_2} \right) \right) + O\left( \frac{1}{\bar{k}_2} \right) ,$$

where

$$\bar{k}_2 = \left\lfloor \frac{k - \sqrt{\beta}(2s - k)}{1 + \sqrt{\beta}} \right\rfloor , \qquad \mu_2 = k/2 - \sqrt{\beta}(2s - k)/2 .$$

Now, we observe that the condition $s \in A$ implies

$$k/2 - \sqrt{\beta}\sqrt{k \log k}/2 \leq \mu_1, \mu_2 \leq k/2 + \sqrt{\beta}\sqrt{k \log k}/2 .$$

and

$$\bar{k}_1, \bar{k}_2 = \Omega(k) .$$

Moreover, from the inequality

$$\frac{x}{\lfloor x/a \rfloor} \left( 1 - \frac{x}{\lfloor x/a \rfloor} \right) \geq a(1 - a)(1 - 1/\sqrt{x})$$

holding for $a \in (1/2, 1)$, and $x \geq \frac{2a}{1-a}$, we also have

$$\frac{\mu_1}{\bar{k}_1} \left( 1 - \frac{\mu_1}{\bar{k}_1} \right) \geq \left( \frac{1}{4} - \frac{\beta}{4} \right) \left( 1 - \frac{1}{\sqrt{\mu_1}} \right) = \left( \frac{1}{4} - \frac{\beta}{4} \right) \left( 1 - O\left( \frac{1}{\sqrt{k}} \right) \right) ,$$

and similarly,

$$\frac{\mu_2}{\bar{k}_2} \left( 1 - \frac{\mu_2}{\bar{k}_2} \right) \geq \left( \frac{1}{4} - \frac{\beta}{4} \right) \left( 1 - \frac{1}{\sqrt{\mu_2}} \right) = \left( \frac{1}{4} - \frac{\beta}{4} \right) \left( 1 - O\left( \frac{1}{\sqrt{k}} \right) \right) .$$

Putting the two entropies together, we then see that

$$\begin{aligned} \mathbb{E}_{h^\star}[H(\alpha_1 \mid s, h^\star)] &= \frac{1}{2} H(\alpha_1 \mid s, h_1) + \frac{1}{2} H(\alpha_1 \mid s, h_2) \\ &\geq \frac{1}{2} \log_2(2\pi e) + \frac{1}{4} \log_2 \left[ k \left( \frac{1}{4} - \frac{\beta}{4} \right) \left( 1 - \tilde{O}\left( \frac{1}{\sqrt{k}} \right) \right) \right] \\ &\quad + \frac{1}{4} \log_2 \left[ k \left( \frac{1}{4} - \frac{\beta}{4} \right) \left( 1 - \tilde{O}\left( \frac{1}{\sqrt{k}} \right) \right) \right] - O\left( \frac{1}{k} \right) \\ &= \frac{1}{2} \log_2 \left( 2\pi e\, k \left( \frac{1}{4} - \frac{\beta}{4} \right) \right) - \tilde{O}\left( \frac{1}{\sqrt{k}} \right) . \end{aligned}$$

Finally, we have

$$H(\alpha_1 \mid \mathcal{B}_1, h^\star) = \underset{s \sim Binom(k,1/2)}{\mathbb{E}} \left[ \mathbb{E}_{h^\star}[H(\alpha_1 \mid s, h^\star)] \right] .$$

Since $\mathbb{P}(\sum_i x_i \in A) \geq 1 - 1/k$ the above gives the lower bound

$$H(\alpha_1 \mid \mathcal{B}_1, h^\star) \geq \left(1 - \frac{1}{k}\right) \frac{1}{2} \log_2 \left(2\pi e\, k \left(\frac{1}{4} - \frac{\beta}{4}\right)\right) - \tilde{O}\left(\frac{1}{\sqrt{k}}\right)$$

as $k$ grows large.

Piecing together and simplifying results in

$$I(h^\star; (\mathcal{B}_1, \alpha_1)) \leq \frac{1}{2} \log_2(\pi e k/2) - \frac{1}{2} \log_2 \left(2\pi e\, k \left(\frac{1}{4} - \frac{\beta}{4}\right)\right) = -\frac{1}{2} \log_2(1 - \beta) + \tilde{O}\left(\frac{1}{\sqrt{k}}\right) ,$$

which is of the form

$$\beta + \tilde{O}\left(\frac{1}{\sqrt{k}}\right)$$

when $\beta \to 0$ and $k$ is large. When $k$ is large enough, the above becomes smaller than $\beta/2$. The lower bound on the number of bags is then of the form

$$m = \Omega\left(\frac{1 - \mathcal{E}(\delta)}{\beta}\right) ,$$

independent of $k$, when $k$ is large enough, and $\beta$ is a small constant (independent of $k$). The sample size $n = mk$ must thus scale linearly with $k$. In particular,

$$n = \Omega\left(\frac{k(1 - \mathcal{E}(\delta))}{\beta}\right) .$$

This concludes the proof.

## B. Proofs and further results for Section 4

This appendix contains the proof of Theorem 4.1, as well as the extension of the ERM estimator in the case where $p$ and $\mathbb{E}[h(x)]$ are unknown.

### B.1. Proof of Theorem 4.1: Known $p$ and $\mathbb{E}[h(x)]$

First, recall the notation introduced in the main body. For bag $z_j = (\mathcal{B}_j, \alpha_j)$, and function $h \in \mathcal{H}$, we set

$$\mathbb{E}_j(h) = \sum_{i=1}^{k} h(x_{j,i}) , \qquad \widetilde{\mathbb{E}}_j(h) = \sum_{i=1}^{k} h(x_{j,i}) - k\mathbb{E}[h(x)] , \qquad \widetilde{\alpha}_j = \alpha_j - p ,$$

where $p = \mathbb{E}[h^\star(x)]$. We also assumed that $\mathbb{E}[h(x)]$, for $h \in \mathcal{H}$, and $p$ are known.

Then, for bag $z_j = (\mathcal{B}_j, \alpha_j)$, recall the definition of *clipped (debiased) square loss*

$$\ell^c(h, z_j) = \underbrace{\frac{1}{k}\left(k\widetilde{\alpha}_j - \widetilde{\mathbb{E}}_j(h)\right)^2 G_j(h)}_{X_j(h)} + \underbrace{\left(\mathbb{E}[h(x)] - p\right)^2}_{a(h)} ,$$

where

$$G_j(h) = \left\{ |k\widetilde{\alpha}_j - \widetilde{\mathbb{E}}_j(h)| \leq \sqrt{8k \log \frac{2}{\theta}} \right\} ,$$

for a suitable value of parameter $\theta > 0$, and then its population version

$$\mathcal{L}_{\mathcal{B}}^c(h) = \mathbb{E}_z \left[\ell^c(h, z)\right] .$$

Also, define the *non-clipped* version of the loss at the bag level:

$$\ell(h, z_j) = \frac{1}{k}\left(k\widetilde{\alpha}_j - \widetilde{\mathbb{E}}_j(h)\right)^2 + \left(\mathbb{E}[h(x)] - p\right)^2 ,$$

and note that its population counterpart $\mathcal{L}_{\mathcal{B}}(h) = \mathbb{E}_z\left[\ell(h, z)\right]$ satisfies

$$\mathcal{L}_{\mathcal{B}}(h) = \mathcal{L}(h) .$$

In order to prove the above claim, set for brevity

$$v_i = y_{j,i} - p + \mathbb{E}[h(x)] - h(x_{j,i}) ,$$

for $j \in [k]$. Then observe that $\ell(h, z_j)$ can be rewritten as $\ell(h, z_j) = \frac{1}{k}\left(\sum_{i=1}^k v_i\right)^2 + (\mathbb{E}[h(x)] - p)^2$ , where the variables $v_i$ are i.i.d. with $\mathbb{E}[v_i] = 0$.

Therefore

$$
\begin{aligned}
\mathcal{L}_{\mathcal{B}}(h) &= \frac{1}{k}\mathbb{E}\left[\left(\sum_{i=1}^k v_i\right)^2\right] + (\mathbb{E}[h(x)] - p)^2 \\
&= \frac{1}{k}\mathbb{E}\left[\sum_{i=1}^k v_i^2\right] + \frac{1}{k}\mathbb{E}\left[\sum_{i\neq j} v_i v_j\right] + (\mathbb{E}[h(x)] - p)^2 \\
&= \mathbb{E}[v_1^2] + (\mathbb{E}[h(x)] - p)^2 \\
&\quad \text{(using the fact that the variables } v_i \text{ are zero-mean and independent)} \\
&= \mathrm{Var}(y_{j,1} - h(x_{j,1})) + (\mathbb{E}[h(x)] - p)^2 \\
&= \mathbb{E}[(y - h(x))^2] \\
&= \mathcal{L}(h) ,
\end{aligned}
$$

as anticipated.

In order to achieve fast rates, we will investigate loss *differences*. This is a by now standard observation that dates back to at least (Massart, 2000; Mendelson, 2002), See also (Bartlett et al., 2005).

Now, for any pair $h_1, h_2 \in \mathcal{H}$, we have

$$
\begin{aligned}
\mathcal{L}_{\mathcal{B}}^c(h_1) - \mathcal{L}_{\mathcal{B}}^c(h_2) &= \mathbb{E}_{z_j}\left[X_j(h_1)G_j(h_1) - X_j(h_2)G_j(h_2) + a(h_1) - a(h_2)\right] \\
&= \mathbb{E}_{z_j}\left[X_j(h_1) - X_j(h_2) + a(h_1) - a(h_2) - X_j(h_1)\overline{G_j(h_1)} + X_j(h_2)\overline{G_j(h_2)}\right] \\
&= \mathbb{E}_{z_j}\left[\ell(h_1, z_j) - \ell(h_2, z_j) - X_j(h_1)\overline{G_j(h_1)} + X_j(h_2)\overline{G_j(h_2)}\right] \\
&= \mathcal{L}(h_1) - \mathcal{L}(h_2) + \mathbb{E}_{z_j}\left[-X_j(h_1)\overline{G_j(h_1)} + X_j(h_2)\overline{G_j(h_2)}\right] ,
\end{aligned}
$$

so that

$$
\begin{aligned}
|\mathcal{L}_{\mathcal{B}}^c(h_1) - \mathcal{L}_{\mathcal{B}}^c(h_2) - (\mathcal{L}(h_1) - \mathcal{L}(h_2))| &\leq \mathbb{E}_{z_j}\left[|X_j(h_1)|\overline{G_j(h_1)} + |X_j(h_2)|\overline{G_j(h_2)}\right] \\
&\leq 4k\,\mathbb{E}_{z_j}\left[\overline{G_j(h_1)} + \overline{G_j(h_2)}\right] \\
&\quad \text{(since } |X_j(h_1)| \text{ and } |X_j(h_1)| \text{ are both bounded by } 4k) \\
&\leq 8k\theta \qquad\qquad\qquad\qquad\qquad\qquad (10) \\
&\quad \text{(since } \mathbb{E}_{z_j}[G_j(h_1)] \text{ and } \mathbb{E}_{z_j}[G_j(h_1)] \text{ are both } \geq 1 - \theta) .
\end{aligned}
$$

Moreover,

$$
\left(\ell^c(h_1, z_j) - \ell^c(h_2, z_j)\right)^2
$$

$$
= \left(X_j(h_1)G_j(h_1) - X_j(h_2)G_j(h_2) + a(h_1) - a(h_2)\right)^2
$$

$$
= \left(\left(X_j(h_1) - X_j(h_2)\right)G_j(h_1)G_j(h_2) + X_j(h_1)G_j(h_1)\overline{G_j(h_2)} - X_j(h_2)\overline{G_j(h_1)}G_j(h_2) + a(h_1) - a(h_2)\right)^2
$$

$$
\leq 4\left(X_j(h_1) - X_j(h_2)\right)^2 G_j(h_1)G_j(h_2) + 4X_j^2(h_1)G_j(h_1)\overline{G_j(h_2)} + 4X_j^2(h_2)\overline{G_j(h_1)}G_j(h_2) + 4\left(a(h_1) - a(h_2)\right)^2
$$

(using $(a + b + c + d)^2 \leq 4(a^2 + b^2 + c^2 + d^2)$) .

Now, define $\gamma(h_1, h_2) = \mathbb{E}[(h_1(x) - h_2(x))^2]$. We have

$$
\left(a(h_1) - a(h_2)\right)^2 = \left(\left(\mathbb{E}[h_1(x)] - p\right)^2 - \left(\mathbb{E}[h_2(x)] - p\right)^2\right)^2
$$

$$
= \left(\mathbb{E}[h_1(x)] - \mathbb{E}[h_2(x)]\right)^2 \left(\mathbb{E}[h_1(x)] + \mathbb{E}[h_2(x)] - 2p\right)^2
$$

$$
= \underbrace{\left(\mathbb{E}[h_1(x) - h_2(x)]\right)^2}_{\leq \mathbb{E}[(h_1(x) - h_2(x))^2] = \gamma(h_1, h_2)} \underbrace{\left(\mathbb{E}[h_1(x)] + \mathbb{E}[h_2(x)] - 2p\right)^2}_{\leq 4}
$$

$$
\leq 4\gamma(h_1, h_2) .
$$

Taking the expectation then gives

$$
\mathbb{E}\left[\left(\ell^c(h_1, z_j) - \ell^c(h_2, z_j)\right)^2\right]
$$

$$
\leq 4\mathbb{E}\left[\left(X_j(h_1) - X_j(h_2)\right)^2 G_j(h_1)G_j(h_2)\right] + 4\mathbb{E}\left[X_j^2(h_1)G_j(h_1)\overline{G_j(h_2)}\right] + 4\mathbb{E}\left[X_j^2(h_2)\overline{G_j(h_1)}G_j(h_2)\right]
$$

$$
+ 16\gamma(h_1, h_2)
$$

$$
\leq 4\mathbb{E}\left[\left(X_j(h_1) - X_j(h_2)\right)^2 G_j(h_1)G_j(h_2)\right] + 128k^2\theta + 16\gamma(h_1, h_2)
$$

$$
= \frac{4}{k^2}\mathbb{E}\left[\left(\widetilde{\mathbb{E}}_j(h_1) - \widetilde{\mathbb{E}}_j(h_2)\right)^2 \left(2k\widetilde{\alpha}_j - \widetilde{\mathbb{E}}_j(h_1) - \widetilde{\mathbb{E}}_j(h_2)\right)^2 G_j(h_1)G_j(h_2)\right] + 128k^2\theta + 16\gamma(h_1, h_2)
$$

$$
\leq \frac{4}{k^2}\mathbb{E}\left[\left(\widetilde{\mathbb{E}}_j(h_1) - \widetilde{\mathbb{E}}_j(h_2)\right)^2 \left(|k\widetilde{\alpha}_j - \widetilde{\mathbb{E}}_j(h_1)| + |k\widetilde{\alpha}_j - \widetilde{\mathbb{E}}_j(h_2)|\right)^2 G_j(h_1)G_j(h_2)\right] + 128k^2\theta + 16\gamma(h_1, h_2)
$$

$$
\leq \frac{128}{k}\mathbb{E}\left[\left(\widetilde{\mathbb{E}}_j(h_1) - \widetilde{\mathbb{E}}_j(h_2)\right)^2\right]\log\frac{2}{\theta} + 128k^2\theta + 16\gamma(h_1, h_2)
$$

$$
= 128\,\mathrm{Var}\left(h_1(x) - h_2(x)\right)\log\frac{2}{\theta} + 128k^2\theta + 16\gamma(h_1, h_2)
$$

$$
\leq 144\,\gamma(h_1, h_2)\log\frac{2}{\theta} + 128k^2\theta . \tag{11}
$$

Further, recalling that $\widehat{h}^\star$ is the best-in-class hypothesis, and $h^\star$ is the Bayes predictor $h^\star(x) = \mathbb{P}(y = 1|x)$, we observe

that, for any $h \in \mathcal{H}$, we can write

$$
\begin{aligned}
\gamma(h, \widehat{h}^\star) &= \mathbb{E}[(h(x) - h^\star(x) + h^\star(x) - \widehat{h}^\star(x))^2] \\
&= \mathbb{E}[(h(x) - h^\star(x))^2] - \mathbb{E}[(\widehat{h}^\star(x) - h^\star(x))^2] + 2\,\mathbb{E}[(h^\star(x) - \widehat{h}^\star(x))(h(x) - \widehat{h}^\star(x))] \\
&= \mathcal{L}(h) - \mathcal{L}(\widehat{h}^\star) + 2\,\mathbb{E}[(h^\star(x) - \widehat{h}^\star(x))(h(x) - \widehat{h}^\star(x))] \\
&\quad\text{(since } \mathcal{L}(h_1) - \mathcal{L}(h_2) = \mathbb{E}[(h_1(x) - h^\star(x))^2] - \mathbb{E}[(h_2(x) - h^\star(x))^2] \text{ for any } h_1, h_2) \\
&\leq \mathcal{L}(h) - \mathcal{L}(\widehat{h}^\star) + 2\sqrt{\mathbb{E}[(h^\star(x) - \widehat{h}^\star(x))^2]}\sqrt{\mathbb{E}[(h(x) - \widehat{h}^\star(x))^2]} \\
&\quad\text{(from the Cauchy-Schwarz inequality)} \\
&= \mathcal{L}(h) - \mathcal{L}(\widehat{h}^\star) + 2\sqrt{\gamma(h^\star, \widehat{h}^\star)}\sqrt{\gamma(h, \widehat{h}^\star)}\,.
\end{aligned}
$$

Solving for $\gamma(h, \widehat{h}^\star)$, and setting for brevity $\Delta\mathcal{L} = \mathcal{L}(h) - \mathcal{L}(\widehat{h}^\star)$, this implies

$$
\gamma(h, \widehat{h}^\star) \leq \Delta\mathcal{L} + 2\gamma(\widehat{h}^\star, h^\star) + 2\sqrt{\gamma^2(\widehat{h}^\star, h^\star) + \gamma(\widehat{h}^\star, h^\star)\Delta\mathcal{L}}
$$

Using the inequality $\sqrt{a + b} \leq \sqrt{a} + b/(2\sqrt{a})$, with $a = \gamma^2(\widehat{h}^\star, h^\star)$ we then have

$$
\gamma(h, \widehat{h}^\star) \leq 4\gamma(\widehat{h}^\star, h^\star) + 2\Delta\mathcal{L}\,.
$$

Combined with (11) this gives, for any $h \in \mathcal{H}$,

$$
\mathbb{E}\left[\left(\ell^c(h, z_j) - \ell^c(\widehat{h}^\star, z_j)\right)^2\right] = O\left(\left(\gamma(\widehat{h}^\star, h^\star) + \mathcal{L}(h) - \mathcal{L}(\widehat{h}^\star)\right)\log\frac{1}{\theta} + k^2\theta\right)\,. \tag{12}
$$

Finally, because of the explicit clipping, we also have, deterministically,

$$
|\ell^c(h_1, z_j) - \ell^c(h_2, z_j)| \leq 8\log\frac{2}{\theta} + 1\,. \tag{13}
$$

Consider now the difference of bag-level clipped losses

$$
\ell^c(h, \widehat{h}^\star; z_j) = \ell^c(h, z_j) - \ell^c(\widehat{h}^\star, z_j)\,,
$$

and define the empirical measure

$$
\widehat{\ell^c}(h, \widehat{h}^\star; S) = \frac{1}{m}\sum_{j=1}^m \ell^c(h, \widehat{h}^\star; z_j) = \frac{1}{m}\sum_{j=1}^m \ell^c(h, z_j) - \frac{1}{m}\sum_{j=1}^m \ell^c(\widehat{h}^\star, z_j)
$$

and the ERM estimator

$$
\widehat{h} = \widehat{h}(S) = \arg\min_{h \in \mathcal{H}} \frac{1}{m}\sum_{j=1}^m \ell^c(h, \widehat{h}^\star; z_j) = \arg\min_{h \in \mathcal{H}} \frac{1}{m}\sum_{j=1}^m \ell^c(h, z_j)\,.
$$

Recall that we denote by $\mathcal{H}_\beta$ the set of $\beta$-suboptimal hypotheses

$$
\mathcal{H}_\beta = \left\{h \in \mathcal{H} : \mathcal{L}(h) \geq \mathcal{L}(\widehat{h}^\star) + \beta\right\}\,.
$$

We introduce the shorthands

$$
\Delta\mathcal{L}_\mathcal{B}^c(h, \widehat{h}^\star) = \mathcal{L}_\mathcal{B}^c(h) - \mathcal{L}_\mathcal{B}^c(\widehat{h}^\star)\,, \qquad \Delta\mathcal{L}_\mathcal{B}(h, \widehat{h}^\star) = \mathcal{L}_\mathcal{B}(h) - \mathcal{L}_\mathcal{B}(\widehat{h}^\star)\,,
$$

where, we recall, $\mathcal{L}_\mathcal{B}^c(h) = \mathbb{E}_{z_j}[\ell^c(h, z_j)]$, and $\mathcal{L}_\mathcal{B}(h) = \mathbb{E}_{z_j}[\ell(h, z_j)]$.

Also recall that $\mathcal{L}_\mathcal{B}(h) = \mathcal{L}(h)$. We can write

$$
\begin{aligned}
\left\{\widehat{h} \in \mathcal{H}_\beta\right\} &\leq \left\{\exists h \in \mathcal{H}_\beta \ : \ \widehat{\ell}^c(h, \widehat{h}^\star; S) \leq \widehat{\ell}^c(\widehat{h}^\star, \widehat{h}^\star; S)\right\} \\
&= \left\{\exists h \in \mathcal{H}_\beta \ : \ \widehat{\ell}^c(h, \widehat{h}^\star; S) - \Delta\mathcal{L}_\mathcal{B}^c(h, \widehat{h}^\star) \leq \widehat{\ell}^c(\widehat{h}^\star, \widehat{h}^\star; S) - \Delta\mathcal{L}_\mathcal{B}^c(h, \widehat{h}^\star)\right\} \\
&\leq \left\{\exists h \in \mathcal{H}_\beta \ : \ \Delta\mathcal{L}_\mathcal{B}^c(h, \widehat{h}^\star) - \widehat{\ell}^c(h, \widehat{h}^\star; S) \geq \Delta\mathcal{L}_\mathcal{B}(h, \widehat{h}^\star) - 8k\theta\right\} \quad (14)
\end{aligned}
$$
(since $\widehat{\ell}^c(\widehat{h}^\star, \widehat{h}^\star; S) = 0$ and, because of (10), $\Delta\mathcal{L}_\mathcal{B}^c(h, \widehat{h}^\star) \geq \Delta\mathcal{L}_\mathcal{B}(h, \widehat{h}^\star) - 8k\theta$).

Since we assumed $|\mathcal{H}| < \infty$, from (14) we can write

$$
\left\{\widehat{h} \in \mathcal{H}_\beta\right\} \leq \sum_{h \in \mathcal{H}_\beta} \left\{\Delta\mathcal{L}_\mathcal{B}^c(h, \widehat{h}^\star) - \widehat{\ell}^c(h, \widehat{h}^\star; S) \geq \Delta\mathcal{L}_\mathcal{B}(h, \widehat{h}^\star) - 8k\theta\right\}.
$$

Our setting

$$
\theta = \frac{\beta}{16k^2}
$$

implies

$$
\Delta\mathcal{L}_\mathcal{B}(h, \widehat{h}^\star) - 8k\theta \geq \Delta\mathcal{L}_\mathcal{B}(h, \widehat{h}^\star) - \beta/(2k) \geq \Delta\mathcal{L}_\mathcal{B}(h, \widehat{h}^\star)/2 \,,
$$

and $k^2\theta = O(\beta) = O(\Delta\mathcal{L}_\mathcal{B}(h, \widehat{h}^\star))$ in the variance bound (12), in both cases using the fact that $h \in \mathcal{H}_\beta$. Moreover, in (13),

$$
|\ell^c(h, \widehat{h}^\star; z_j)| = O\left(\log \frac{k}{\beta}\right) . \quad (15)
$$

From (12) and (15), using the standard Bernstein inequality to each individual $h \in \mathcal{H}_\beta$, and setting for brevity $\mu = \mu(h) = \Delta\mathcal{L}_\mathcal{B}(h, \widehat{h}^\star)$, we can write

$$
\begin{aligned}
\mathbb{P}\left(\widehat{h} \in \mathcal{H}_\beta\right) &\leq \sum_{h \in \mathcal{H}_\beta} \mathbb{P}\left(\Delta\mathcal{L}_\mathcal{B}^c(h, \widehat{h}^\star) - \widehat{\ell}^c(h, \widehat{h}^\star; S) \geq \mu/2\right) \\
&\leq \sum_{h \in \mathcal{H}_\beta} \exp\left(-\frac{m^2\mu^2/8}{m\,O\left(\left(\gamma(\widehat{h}^\star, h^\star) + \mu\right)\log \frac{k}{\beta}\right) + m\,O\left(\mu \log \frac{k}{\beta}\right)}\right) \\
&= \sum_{h \in \mathcal{H}_\beta} \exp\left(-\frac{m\mu^2/8}{O\left(\left(\gamma(\widehat{h}^\star, h^\star) + \mu\right)\log \frac{k}{\beta}\right)}\right) .
\end{aligned}
$$

Since the function $\mu \to \frac{\mu^2}{\gamma+\mu}$ is increasing in $\mu \geq 0$ for any $\gamma \geq 0$, and in our case $\mu \geq \beta$ for any $h \in \mathcal{H}_\beta$, a lower bound on the fraction in the exponential is simply obtained by replacing $\mu$ with $\beta$. This yields

$$
\mathbb{P}\left(\widehat{h} \in \mathcal{H}_\beta\right) \leq |\mathcal{H}_\beta| \exp\left(-\frac{m\beta^2/8}{O\left(\left(\gamma(\widehat{h}^\star, h^\star) + \beta\right)\log \frac{k}{\beta}\right)}\right) .
$$

If we want the right-hand side to be less than $\delta$ it then suffices to have

$$
m = O\left(\frac{\left(\gamma(\widehat{h}^\star, h^\star) + \beta\right)\log \frac{k}{\beta}}{\beta^2}\, \log \frac{|\mathcal{H}_\beta|}{\delta}\right) .
$$

In particular, in the realizable case, we have $\gamma(\widehat{h}^\star, h^\star) = 0$, yielding the bound

$$
m = O\left(\frac{\log \frac{k}{\beta}}{\beta}\, \log \frac{|\mathcal{H}_\beta|}{\delta}\right) .
$$

This concludes the proof.

**B.2. Proof of Theorem 4.1: Unknown $p$ and $\mathbb{E}[h(x)]$**

When $p$ and $\mathbb{E}[h(x)]$ are unknown, we simply split the $m$ bags into $m_1 + m_2$ bags, and hence the dataset into two parts:

$$S = \underbrace{((x_{1,1}, \ldots, x_{1,k}), \alpha_1), \ldots, ((x_{m_1,1}, \ldots, x_{m_1,k}), \alpha_{m_1})}_{S_1},$$

$$\underbrace{((x_{m_1+1,1}, \ldots, x_{m_1+1,k}), \alpha_{m_1+1}), \ldots, ((x_{m_1+m_2,1}, \ldots, x_{m_1+m_2,k}), \alpha_{m_1+m_2})}_{S_2},$$

then estimate $p$ and $\mathbb{E}[h(x)]$ on the second half $S_2$. Specifically,

$$\widehat{p} = \frac{1}{m_2} \sum_{j=m_1+1}^{m_1+m_2} \alpha_j$$

and

$$\widehat{\mathbb{E}}[h(x)] = \frac{1}{m_2 k} \sum_{j=m_1+1}^{m_1+m_2} \sum_{i=1}^{k} h(x_{j,i}) .$$

Then, for $j \in [m_1]$, set

$$\mathbb{E}_j(h) = \sum_{i=1}^{k} h(x_{j,i}) , \quad \widetilde{\mathbb{E}}_j(h) = \mathbb{E}_j(h) - k\widehat{\mathbb{E}}[h(x)] , \quad \widetilde{\alpha}_j = \alpha_j - \widehat{p}$$

and re-define the *clipped* bag-level square loss as

$$\ell^c(h, z_j) = \underbrace{\frac{1}{k}\left(k\widetilde{\alpha}_j - \widetilde{\mathbb{E}}_j(h)\right)^2 G_j(h)}_{\widehat{X}_j(h)} + \underbrace{\left(\widehat{\mathbb{E}}[h(x)] - \widehat{p}\right)^2}_{\widehat{a}(h)} ,$$

where now

$$G_j(h) = \left\{ |k\widetilde{\alpha}_j - \widetilde{\mathbb{E}}_j(h)| \leq \sqrt{18k \log \frac{6}{\theta}} \right\} ,$$

for some value of parameter $\theta > 0$ that will be determined later on.

Note that

$$
\begin{aligned}
|k\widetilde{\alpha}_j - \widetilde{\mathbb{E}}_j(h)| &= |k\alpha_j - k\widehat{p} - \mathbb{E}_j(h) + k\widehat{\mathbb{E}}[h(x)]| \\
&= |k(\alpha_j - p) + k(p - \widehat{p}) + k\,\mathbb{E}[h(x)] - \mathbb{E}_j(h) + k(\widehat{\mathbb{E}}[h(x)] - \mathbb{E}[h(x)])| \\
&\leq k|\alpha_j - p + k\,\mathbb{E}[h(x)] - \mathbb{E}_j(h)| + k|p - \widehat{p}| + k|\widehat{\mathbb{E}}[h(x)] - \mathbb{E}[h(x)]| .
\end{aligned}
$$

Hence, for all $t \geq 0$, and any fixed $h \in \mathcal{H}$, and $j \in [m_1]$,

$$\left\{ |k\widetilde{\alpha}_j - \widetilde{\mathbb{E}}_j(h)| \geq t \right\} \leq \left\{ k|\alpha_j - p + k\,\mathbb{E}[h(x)] - \mathbb{E}_j(h)| \geq t_1 \right\} + \left\{ k|p - \widehat{p}| \geq t_2 \right\} + \left\{ k|\widehat{\mathbb{E}}[h(x)] - \mathbb{E}[h(x)]| \geq t_3 \right\} ,$$

where $t_1 + t_2 + t_3 = t$. We set

$$t_1 = \sqrt{8k \log \frac{6}{\theta}} , \quad t_2 = t_3 = \sqrt{\frac{k}{2m_2} \log \frac{6}{\theta}}$$

to conclude from the standard Hoeffding's inequality that, for fixed $j \in [m_1]$,

$$\mathbb{E}\left[ \left\{ k|\alpha_j - p + k\,\mathbb{E}[h(x)] - \mathbb{E}_j(h)| \geq t_1 \right\} \right] \leq \frac{\theta}{3}$$

and, taking expectation over the generation of $S_2$,

$$\mathbb{E}\left[ \left\{ k|p - \widehat{p}| \geq t_2 \right\} \right] \leq \frac{\theta}{3} , \qquad \mathbb{E}\left[ \left\{ k|\widehat{\mathbb{E}}[h(x)] - \mathbb{E}[h(x)]| \geq t_3 \right\} \right] \leq \frac{\theta}{3} .$$

Since $t_1 + t_2 + t_3 \leq \sqrt{18k \log \frac{6}{\theta}}$, this implies

$$\mathbb{E}[G_j(h)] \leq \theta \,.$$

for each individual $h \in \mathcal{H}$, and $j \in [m_1]$.

We now need to proceed with a bias-variance tradeoff, which is similar to, but a bit more involved than the one contained in Section B.1.

We still define the unclipped version of the loss at the bag level as in Section B.1, but now we need to factor in the extra bias coming from the estimators from $S_2$. In particular, for any pair $h_1, h_2 \in \mathcal{H}$, we have

$$\begin{aligned}
\mathcal{L}_{\mathcal{B}}^c(h_1) - \mathcal{L}_{\mathcal{B}}^c(h_2) &= \mathbb{E}_{z_j}\left[\widehat{X}_j(h_1)G_j(h_1) - \widehat{X}_j(h_2)G_j(h_2) + \widehat{a}(h_1) - \widehat{a}(h_2)\right] \\
&= \mathbb{E}_{z_j}\left[\widehat{X}_j(h_1) - \widehat{X}_j(h_2) + \widehat{a}(h_1) - \widehat{a}(h_2) - \widehat{X}_j(h_1)\overline{G_j(h_1)} + \widehat{X}_j(h_2)\overline{G_j(h_2)}\right] \\
&= \mathbb{E}_{z_j}\Big[\widehat{X}_j(h_1) - X_j(h_1) + X_j(h_1) - X_j(h_2) + X_j(h_2) - \widehat{X}_j(h_2) \\
&\qquad\qquad + \widehat{a}(h_1) - a(h_1) + a(h_1) - a(h_2) + a(h_2) - \widehat{a}(h_2) - \widehat{X}_j(h_1)\overline{G_j(h_1)} + \widehat{X}_j(h_2)\overline{G_j(h_2)}\Big] \\
&= \mathbb{E}_{z_j}\left[\ell(h_1, z_j) - \ell(h_2, z_j) - \widehat{X}_j(h_1)\overline{G_j(h_1)} + \widehat{X}_j(h_2)\overline{G_j(h_2)}\right] \\
&\qquad + \mathbb{E}_{z_j}\left[\widehat{X}_j(h_1) - X_j(h_1) + X_j(h_2) - \widehat{X}_j(h_2) + \widehat{a}(h_1) - a(h_1) + a(h_2) - \widehat{a}(h_2)\right] \\
&= \mathcal{L}(h_1) - \mathcal{L}(h_2) + \mathbb{E}_{z_j}\left[-\widehat{X}_j(h_1)\overline{G_j(h_1)} + \widehat{X}_j(h_2)\overline{G_j(h_2)}\right] \\
&\qquad + \mathbb{E}_{z_j}\left[\Delta\widehat{X}_j(h_1, h_2) - \Delta X_j(h_1, h_2) + \Delta\widehat{a}(h_1, h_2) - \Delta a(h_1, h_2)\right]\,,
\end{aligned}$$

where we set for brevity

$$\begin{aligned}
\Delta\widehat{X}_j(h_1, h_2) &= \widehat{X}_j(h_1) - \widehat{X}_j(h_2)\,, & \Delta X_j(h_1, h_2) &= X_j(h_1) - \widehat{X}_j(h_2)\,, \\
\Delta\widehat{a}(h_1, h_2) &= \widehat{a}(h_1) - \widehat{a}(h_2)\,, & \Delta a(h_1, h_2) &= a(h_1) - a(h_2)\,.
\end{aligned}$$

Thus

$$\begin{aligned}
&|\mathcal{L}_{\mathcal{B}}^c(h_1) - \mathcal{L}_{\mathcal{B}}^c(h_2) - (\mathcal{L}(h_1) - \mathcal{L}(h_2))| \\
&\leq \mathbb{E}_{z_j}\left[|\widehat{X}_j(h_1)|\overline{G_j(h_1)} + |\widehat{X}_j(h_2)|\overline{G_j(h_2)}\right] \\
&\qquad + \mathbb{E}_{z_j}\left[|\Delta\widehat{X}_j(h_1, h_2) - \Delta X_j(h_1, h_2)|\right] + \mathbb{E}_{z_j}\left[|\Delta\widehat{a}(h_1, h_2) - \Delta a(h_1, h_2)|\right]\,.
\end{aligned} \tag{16}$$

Now, as in Section B.1,

$$\mathbb{E}_{z_j}\left[|\widehat{X}_j(h_1)|\overline{G_j(h_1)} + |\widehat{X}_j(h_2)|\overline{G_j(h_2)}\right] \leq 8k\theta \,. \tag{17}$$

Moreover, for any $h_1, h_2$,

$$\begin{aligned}
\Delta\widehat{X}_j(h_1, h_2) &= \frac{1}{k}\Big(\mathbb{E}_j(h_2) - \mathbb{E}_j(h_1)\Big)\Big(2k\alpha_j - 2k\widehat{p} - \mathbb{E}_j(h_1) - \mathbb{E}_j(h_2) + k\widehat{\mathbb{E}}[h_1(x)] + k\widehat{\mathbb{E}}[h_2(x)]\Big) \\
\Delta X_j(h_1, h_2) &= \frac{1}{k}\Big(\mathbb{E}_j(h_2) - \mathbb{E}_j(h_1)\Big)\Big(2k\alpha_j - 2kp - \mathbb{E}_j(h_1) - \mathbb{E}_j(h_2) + k\mathbb{E}[h_1(x)] + k\mathbb{E}[h_2(x)]\Big)\,,
\end{aligned}$$

so that

$$\begin{aligned}
&\Delta\widehat{X}_j(h_1, h_2) - \Delta X_j(h_1, h_2) \\
&\qquad = \frac{1}{k}\Big(\mathbb{E}_j(h_2) - \mathbb{E}_j(h_1)\Big)\Big(2k(p - \widehat{p}) + k(\widehat{\mathbb{E}}[h_1(x)] - \mathbb{E}[h_1(x)]) + k(\widehat{\mathbb{E}}[h_2(x)] - \mathbb{E}[h_2(x)])\Big)
\end{aligned}$$

and

$$|\Delta \widehat{X}_j(h_1, h_2) - \Delta X_j(h_1, h_2)|$$
$$\leq \left| \mathbb{E}_j(h_2) - \mathbb{E}_j(h_1) \right| \left( 2|p - \widehat{p}| + \left| \widehat{\mathbb{E}}[h_1(x)] - \mathbb{E}[h_1(x)] \right| + \left| \widehat{\mathbb{E}}[h_2(x)] - \mathbb{E}[h_2(x)] \right| \right). \tag{18}$$

From Hoeffding's inequality we know that the sum of the three absolute values in the second factor is overall bounded by

$$\sqrt{\frac{8}{m_2 k} \log \frac{6}{\delta}}$$

with probability $\geq 1 - \delta$ over the random draw of $S_2$. On the other hand, by convexity,

$$\mathbb{E}_{z_j}\left[ \left| \mathbb{E}_j(h_2) - \mathbb{E}_j(h_1) \right| \right] \leq \sqrt{\mathbb{E}_{z_j}\left[ \left( \mathbb{E}_j(h_2) - \mathbb{E}_j(h_1) \right)^2 \right]} = \sqrt{k\gamma(h_1, h_2)}.$$

Similarly, if we set for short

$$\Delta \widehat{\mathbb{E}} = \widehat{\mathbb{E}}[h_2(x)] - \widehat{\mathbb{E}}[h_1(x)], \qquad \Delta \mathbb{E} = \mathbb{E}[h_2(x)] - \mathbb{E}[h_1(x)]$$

we can write

$$\Delta \widehat{a}(h_1, h_2) = (\Delta \widehat{\mathbb{E}} - \Delta \mathbb{E})\left( 2\widehat{p} - \widehat{\mathbb{E}}[h_1(x)] - \widehat{\mathbb{E}}[h_2(x)] \right) + \Delta \mathbb{E}\left( 2\widehat{p} - \widehat{\mathbb{E}}[h_1(x)] - \widehat{\mathbb{E}}[h_2(x)] \right)$$
$$\Delta a(h_1, h_2) = \Delta \mathbb{E}\left( 2\widehat{p} - \mathbb{E}[h_1(x)] - \mathbb{E}[h_2(x)] \right)$$

Hence

$$\Delta \widehat{a}(h_1, h_2) - \Delta a(h_1, h_2)$$
$$= (\Delta \widehat{\mathbb{E}} - \Delta \mathbb{E})\left( 2\widehat{p} - \widehat{\mathbb{E}}[h_1(x)] - \widehat{\mathbb{E}}[h_2(x)] \right) + \Delta \mathbb{E}\left( 2(\widehat{p} - p) + (\mathbb{E}[h_1(x)] - \widehat{\mathbb{E}}[h_1(x)]) + (\mathbb{E}[h_2(x)] - \widehat{\mathbb{E}}[h_2(x)]) \right)$$

which implies

$$|\Delta \widehat{a}(h_1, h_2) - \Delta a(h_1, h_2)|$$
$$\leq |\Delta \widehat{\mathbb{E}} - \Delta \mathbb{E}| \underbrace{\left| 2\widehat{p} - \widehat{\mathbb{E}}[h_1(x)] - \widehat{\mathbb{E}}[h_2(x)] \right|}_{\leq 2} + |\Delta \mathbb{E}|\left( 2|\widehat{p} - p| + |\mathbb{E}[h_1(x)] - \widehat{\mathbb{E}}[h_1(x)]| + |\mathbb{E}[h_2(x)] - \widehat{\mathbb{E}}[h_2(x)]| \right)$$
$$\leq 2|\Delta \widehat{\mathbb{E}} - \Delta \mathbb{E}| + \sqrt{\gamma(h_1, h_2)}\left( 2|\widehat{p} - p| + |\mathbb{E}[h_1(x)] - \widehat{\mathbb{E}}[h_1(x)]| + |\mathbb{E}[h_2(x)] - \widehat{\mathbb{E}}[h_2(x)]| \right), \tag{19}$$

where in the last step we used the fact that

$$|\Delta \mathbb{E}| \leq \mathbb{E}[|h_1(x) - h_2(x)|] \leq \sqrt{\gamma(h_1, h_2)}.$$

As before, Hoeffding's inequality allows us to see that the sum of the three absolute values in the second term is overall bounded by

$$\sqrt{\frac{8}{m_2 k} \log \frac{6}{\delta}}$$

with probability $\geq 1 - \delta$ over the random draw of $S_2$. Moreover, since $\mathrm{Var}\left( h_2(x) - h_1(x) \right) \leq \gamma(h_1, h_2)$, Bernstein's inequality shows that with the same probability we also have

$$|\Delta \widehat{\mathbb{E}} - \Delta \mathbb{E}| \leq \sqrt{\frac{2\gamma(h_1, h_2)}{m_2 k} \log \frac{1}{\delta}} + \frac{2}{3m_2 k} \log \frac{1}{\delta}.$$

We piece together as in (16), and take a union bound over $h_1, h_2 \in \mathcal{H}$. We conclude that with probability $\geq 1 - \delta$ over $S_2$,

$$|\mathcal{L}_{\mathcal{B}}^c(h_1) - \mathcal{L}_{\mathcal{B}}^c(h_2) - (\mathcal{L}(h_1) - \mathcal{L}(h_2))| \leq 8k\theta + O\left( \sqrt{\frac{\gamma(h_1, h_2)}{m_2} \log \frac{|\mathcal{H}|}{\delta}} + \frac{1}{m_2 k} \log \frac{|\mathcal{H}|}{\delta} \right), \tag{20}$$

which is the counterpart of (10).

As in Section B.1, we continue by bounding the second moment. We have

$$
\left( \ell^c(h_1, z_j) - \ell^c(h_2, z_j) \right)^2
$$

$$
= \left( \widehat{X}_j(h_1) G_j(h_1) - \widehat{X}_j(h_2) G_j(h_2) + \widehat{a}(h_1) - \widehat{a}(h_2) \right)^2
$$

$$
= \Bigg( \widehat{X}_j(h_1) - X_j(h_1) + X_j(h_1) - X_j(h_2) + X_j(h_2) - \widehat{X}_j(h_2)
$$

$$
+ \widehat{a}(h_1) - a(h_1) + a(h_1) - a(h_2) + a(h_2) - \widehat{a}(h_2) - \widehat{X}_j(h_1)\overline{G_j(h_1)} + \widehat{X}_j(h_2)\overline{G_j(h_2)} \Bigg)^2
$$

$$
= \Bigg( X_j(h_1) + a(h_1) - X_j(h_2) - a(h_2) + \Delta\widehat{X}_j(h_1, h_2) - \Delta X_j(h_1, h_2) + \Delta\widehat{a}(h_1, h_2) - \Delta a(h_1, h_2)
$$

$$
- \widehat{X}_j(h_1)\overline{G_j(h_1)} + \widehat{X}_j(h_2)\overline{G_j(h_2)} \Bigg)^2
$$

$$
\leq 4 \underbrace{\left( X_j(h_1) + a(h_1) - X_j(h_2) - a(h_2) \right)^2}_{(I)} + 4 \underbrace{\left( \Delta\widehat{X}_j(h_1, h_2) - \Delta X_j(h_1, h_2) \right)^2}_{(II)} + 4 \underbrace{\left( \Delta\widehat{a}(h_1, h_2) - \Delta a(h_1, h_2) \right)^2}_{(III)}
$$

$$
+ 4 \underbrace{\left( \widehat{X}_j(h_1)\overline{G_j(h_1)} + \widehat{X}_j(h_2)\overline{G_j(h_2)} \right)^2}_{(IV)}
$$

(using $(a + b + c + d)^2 \leq 4(a^2 + b^2 + c^2 + d^2)$) .

At this point, we take expectation w.r.t. $z_j$, and use past calculations. In particular, from (11)

$$
\mathbb{E}_{z_j}[(I)] \leq 144\,\gamma(h_1, h_2)\, \log\frac{2}{\theta} + 128k^2\theta \ .
$$

Moreover, from the arguments surrounding Eq. (18), with probability at least $1 - \delta$ over $S_2$,

$$
\mathbb{E}_{z_j}[(II)] \leq \frac{8\gamma(h_1, h_2)}{m_2}\, \log\frac{6}{\delta} \ .
$$

Likewise, from the arguments surrounding Eq. (19), with probability at least $1 - \delta$ over $S_2$,

$$
\mathbb{E}_{z_j}[(III)] \leq \frac{16\gamma(h_1, h_2)}{m_2\, k}\, \log\frac{1}{\delta} + 2\left( \frac{2}{3m_2\, k}\, \log\frac{1}{\delta} \right)^2 + \frac{16\gamma(h_1, h_2)}{m_2\, k}\, \log\frac{6}{\delta}
$$

$$
= O\left( \frac{\gamma(h_1, h_2)}{m_2\, k}\, \log\frac{1}{\delta} + \left( \frac{1}{m_2\, k}\, \log\frac{1}{\delta} \right)^2 \right) ,
$$

where we repeatedly used the upper bound $(a + b)^2 \leq 2(a^2 + b^2)$.

Finally, from the same argument as in (17),

$$
\mathbb{E}_{z_j}[(IV)] \leq \mathbb{E}_{z_j}\left[ \left( \widehat{X}_j(h_1)\overline{G_j(h_1)} + \widehat{X}_j(h_2)\overline{G_j(h_2)} \right)^2 \right]
$$

$$
\leq 2\mathbb{E}_{z_j}\left[ \left( \widehat{X}_j(h_1) \right)^2 \overline{G_j(h_1)} + \left( \widehat{X}_j(h_2) \right)^2 \overline{G_j(h_2)} \right]
$$

$$
\leq 64k^2\theta \ .
$$

Combining the above terms, and taking a union bound over $h_1, h_2 \in \mathcal{H}$, we have shown that, with probability at least $1 - \delta$ over $S_2$,

$$\mathbb{E}_{z_j}\left[\left(\ell^c(h_1, z_j) - \ell^c(h_2, z_j)\right)^2\right] = O\left(\gamma(h_1, h_2)\log\frac{1}{\theta} + k^2\theta + \frac{\gamma(h_1, h_2)}{m_2\,k}\log\frac{1}{\delta} + \left(\frac{1}{m_2\,k}\log\frac{1}{\delta}\right)^2\right),$$

which is the counterpart of (11).

As for the counterpart of (13), we also have deterministically

$$|\ell^c(h_1, z_j) - \ell^c(h_2, z_j)| \leq 18\log\frac{6}{\theta} + 1.$$

From this point on, we follow the proof of Section B.1 by setting $h_1 = \widehat{h}$, and $h_2 = \widehat{h}^\star$. Recall that for all $h \in \mathcal{H}_\beta$, the quantity $\gamma(h, \widehat{h}^\star)$ can be upper bounded as

$$\gamma(h, \widehat{h}^\star) \leq 4\gamma^* + 2\Delta\mathcal{L}(h, \widehat{h}^\star),$$

where $\gamma^* = \gamma(\widehat{h}^\star, h^\star)$, and $\Delta\mathcal{L}(h, \widehat{h}^\star) = \mathcal{L}(h) - \mathcal{L}(\widehat{h}^\star)$. We then set

$$\theta = \frac{\beta}{16k^2}$$

as in Section B.1. This implies

$$|\mathcal{L}_\mathcal{B}^c(h) - \mathcal{L}_\mathcal{B}^c(\widehat{h}^\star) - (\mathcal{L}(h) - \mathcal{L}(\widehat{h}^\star))| \leq \frac{\beta}{2k} + O\left(\sqrt{\frac{\gamma^* + \Delta\mathcal{L}(h, \widehat{h}^\star)}{m_2}\log\frac{|\mathcal{H}|}{\delta}} + \frac{1}{m_2\,k}\log\frac{|\mathcal{H}|}{\delta}\right),$$

and

$$\mathbb{E}_{z_j}\left[\left(\ell^c(h_1, z_j) - \ell^c(h_2, z_j)\right)^2\right] = O\left((\gamma^* + \Delta\mathcal{L}(h, \widehat{h}^\star))\log\frac{k}{\beta} + \frac{(\gamma^* + \Delta\mathcal{L}(h, \widehat{h}^\star))}{m_2\,k}\log\frac{1}{\delta} + \left(\frac{1}{m_2\,k}\log\frac{1}{\delta}\right)^2\right),$$

and

$$|\ell^c(h_1, z_j) - \ell^c(h_2, z_j)| = O\left(\log\frac{k}{\beta}\right),$$

holding with probability $\geq 1 - \delta$ over the generation of $S_2$. Now if, for all $h \in \mathcal{H}_\beta$,

$$m_2 = \Omega\left(\frac{\gamma^* + \Delta\mathcal{L}(h, \widehat{h}^\star)}{(\Delta\mathcal{L}(h, \widehat{h}^\star))^2}\log\frac{|\mathcal{H}|}{\delta}\right)$$

for appropriate constants hidden in the $\Omega$-notation, we see that the bias bound satisfies

$$\frac{\beta}{2k} + O\left(\sqrt{\frac{\gamma^* + \Delta\mathcal{L}(h, \widehat{h}^\star)}{m_2}\log\frac{|\mathcal{H}|}{\delta}} + \frac{1}{m_2\,k}\log\frac{|\mathcal{H}|}{\delta}\right) \leq \frac{\beta}{2} + \frac{\Delta\mathcal{L}(h, \widehat{h}^\star)}{4} \leq \frac{3}{4}\Delta\mathcal{L}(h, \widehat{h}^\star),$$

so that, when $h \in \mathcal{H}_\beta$,

$$\Delta\mathcal{L}_\mathcal{B}(h, \widehat{h}^\star) - \frac{\beta}{2k} + O\left(\sqrt{\frac{\gamma^* + \Delta\mathcal{L}(h, \widehat{h}^\star)}{m_2}\log\frac{|\mathcal{H}|}{\delta}} + \frac{1}{m_2\,k}\log\frac{|\mathcal{H}|}{\delta}\right) \geq \frac{1}{4}\Delta\mathcal{L}_\mathcal{B}(h, \widehat{h}^\star) = \frac{1}{4}\left(\mathcal{L}(h) - \mathcal{L}(\widehat{h}^\star)\right).$$

Moreover, under the same condition on $m_2$,

$$\mathbb{E}_{z_j}\left[\left(\ell^c(h_1, z_j) - \ell^c(h_2, z_j)\right)^2\right] = O\left((\gamma^* + \Delta\mathcal{L}_\mathcal{B}(h, \widehat{h}^\star))\log\frac{k}{\beta} + (\Delta\mathcal{L}(h, \widehat{h}^\star))^2\right).$$

We proceed as in Section B.1 with an application of Bernstein's inequality on the generation of $S_1$. We set for brevity $\mu = \mu(h) = \Delta\mathcal{L}_{\mathcal{B}}(h, \widehat{h}^\star)$. We can write

$$\mathbb{P}\left(\widehat{h} \in \mathcal{H}_\beta\right) \leq \sum_{h \in \mathcal{H}_\beta} \mathbb{P}\left(\Delta\mathcal{L}_{\mathcal{B}}^c(h, \widehat{h}^\star) - \widehat{\ell}^c(h, \widehat{h}^\star; S) \geq \mu/4\right)$$

$$\leq \sum_{h \in \mathcal{H}_\beta} \exp\left(-\frac{m_1^2\,\mu^2/32}{m_1\,O\left((\gamma^* + \mu)\log\frac{k}{\beta} + \mu^2\right) + m_1\,O\left(\mu\log\frac{k}{\beta}\right)}\right)$$

$$= \sum_{h \in \mathcal{H}_\beta} \exp\left(-\frac{m_1\,\mu^2/32}{O\left((\gamma^* + \mu)\log\frac{k}{\beta} + \mu^2\right)}\right).$$

This time we use the fact that the function $\mu \to \frac{\mu^2}{\gamma A + \mu A + \mu^2}$ is increasing in $\mu \geq 0$ for any $A, \gamma \geq 0$. Since $\mu \geq \beta$, this gives

$$\mathbb{P}\left(\widehat{h} \in \mathcal{H}_\beta\right) \leq |\mathcal{H}_\beta|\,\exp\left(-\frac{m_1\,\beta^2/32}{O\left((\gamma^* + \beta)\log\frac{k}{\beta} + \beta^2\right)}\right) = |\mathcal{H}_\beta|\,\exp\left(-\frac{m_1\,\beta^2/32}{O\left((\gamma^* + \beta)\log\frac{k}{\beta}\right)}\right).$$

Making the right-hand side smaller than $\delta$ gives

$$m_1 = O\left(\frac{(\gamma^* + \beta)\log\frac{k}{\beta}}{\beta^2}\log\frac{|\mathcal{H}_\beta|}{\delta}\right).$$

Since the condition on $m_2$ is satisfied for all $h \in \mathcal{H}_\beta$ by selecting

$$m_2 = O\left(\frac{\gamma^* + \beta}{\beta^2}\log\frac{|\mathcal{H}|}{\delta}\right),$$

the overall sample complexity is

$$m = m_1 + m_2 = O\left(\frac{(\gamma^* + \beta)\log\frac{k}{\beta}}{\beta^2}\log\frac{|\mathcal{H}|}{\delta}\right).$$

This concludes the proof.

*Remark* B.1. Despite not shown in the above proof, one can easily see that the dependence on $|\mathcal{H}|$ in $m_2$ can be replaced by a dependence on $|\mathcal{H}_\beta|$, thus the overall sample complexity $m_1 + m_2$ is indeed only depending on $|\mathcal{H}_\beta|$.

## C. Proofs for Section 5

In this section we detail the proof of Theorem 5.1. For the analysis, we require the definition of a smooth function along with some basic properties that we recall now. A function $f : \mathbb{R}^d \to \mathbb{R}$ is said to be $\zeta$-smooth (with respect to the Euclidean norm $\|\cdot\|$) if its gradient is $\zeta$-Lipschitz, namely if

$$\forall\, x, y \in \mathbb{R}^d : \qquad \|\nabla f(x) - \nabla f(y)\| \leq \zeta\|x - y\|.$$

For a twice continuously-differentiable convex function $f$, a sufficient condition for $\zeta$-smoothness is that $0 \preceq \nabla^2 f(x) \preceq \zeta I$ for all $x \in \mathbb{R}^d$. Finally, a smooth function satisfies the so-called descent lemma, from which it follows that

$$\forall\, x \in \mathbb{R}^d : \qquad \|\nabla f(x)\|^2 \leq f(x) - f(x^*),$$

where $x^* \in \mathbb{R}^d$ is a global minimizer of $f$.

For the sake of this section, let us introduce some short-hand notation:

$$\widetilde{\ell}_j = \widetilde{\ell}(\cdot, z_j)$$

and

$$\delta = \frac{\rho_w^2 \rho_x^2}{9km(\rho_w \rho_x + \rho_y)^2} \ .$$

We begin the analysis by establishing that the losses $\widetilde{\ell}_j$ underlying Algorithm 2 are indeed convex and smooth.

**Lemma C.1.** *For all $j$, the loss $\widetilde{\ell}_j$ is convex, nonnegative and $\zeta$-smooth, with $\zeta = (k\theta^2 + 1)\rho_x^2$.*

*Proof.* Convexity and nonnegativity are immediate. To see the claim about smoothness, note that $\widetilde{\ell}_j$ is either identically zero if $\|\bar{x}_j - \mu_x\| > \theta\rho_x$, in which case it is smooth, or is identical to the function $\ell(\cdot, z_j)$ whenever $\|\bar{x}_j - \mu_x\| \leq \theta\rho_x$. In the latter case, the Hessian is

$$\nabla^2 \widetilde{\ell}_j(w) = \nabla^2 \ell(w, z_j) = k(\bar{x}_j - \mu_x)(\bar{x}_j - \mu_x)^T + \mu_x \mu_x^T \ ,$$

whose eigenvalues are upper bounded by $k\|\bar{x}_j - \mu_x\|^2 + \|\mu_x\|^2 \leq (k\theta^2 + 1)\rho_x^2$ . $\qquad\square$

Next, we demonstrate that in expectation, the truncated losses $\widetilde{\ell}_j$ are nearly unbiased estimates of the population loss $\mathcal{L}$. The bias in this estimation is controlled by the truncation threshold $\theta$.

**Lemma C.2.** *Suppose $\theta \geq \sqrt{8\log(1/\delta)/k}$. Then for all $j$ and $w$ such that $\|w\| \leq \rho_w$, one has*

$$-9k(\rho_w \rho_x + \rho_y)^2 \delta \leq \mathbb{E}[\widetilde{\ell}_j(w)] - \mathcal{L}(w) \leq 0 \ .$$

*Proof.* First, we note that $\mathbb{E}[\ell_j(w)] = \mathcal{L}(w)$; this follows directly from Equation (3). Further, for all $w$ such that $\|w\| \leq \rho_w$ we have $\ell_j(w) \leq 9k(\rho_w \rho_x + \rho_y)^2$, hence

$$\mathbb{E}[\ell_j(w)] - \mathbb{E}[\widetilde{\ell}_j(w)] = \mathbb{E}\Big[\ell_j(w) \cdot \{\|\bar{x}_j - \mu_x\| > \theta\rho_x\}\Big]$$
$$\leq 9k(\rho_w \rho_x + \rho_y)^2 \, \mathbb{P}\big(\|\bar{x}_j - \mu_x\| > \theta\rho_x\big) \ .$$

To bound the probability on the right-hand side, note that $\bar{x}_j - \mu_x$ is an average of $k$ i.i.d. zero-mean random vectors bounded by $\rho_x$; from a vector Hoeffding bound (e.g., Boucheron et al., 2013, Example 6.3) it is straightforward to derive that for any $\epsilon \geq 2\rho_x/\sqrt{k}$ ,

$$\mathbb{P}\Big(\|\bar{x}_j - \mu_x\| > \epsilon\Big) \leq \exp\left(-\frac{k\epsilon^2}{8\rho_x^2}\right).$$

Thus, for any $\theta \geq \sqrt{8\log(1/\delta)/k}$,

$$\mathbb{P}\Big(\|\bar{x}_j - \mu_x\| > \theta\rho_x\Big) \leq \exp\big(-\tfrac{1}{8}k\theta^2\big) \leq \delta,$$

and therefore

$$\mathbb{E}[\ell_j(w)] - \mathbb{E}[\widetilde{\ell}_j(w)] \leq 9k(\rho_w \rho_x + \rho_y)^2 \delta,$$

which concludes the proof. $\qquad\square$

Henceforth, we let $\zeta = (k\theta^2 + 1)\rho_x^2$ and fix $\theta = \sqrt{8\log(1/\delta)/k}$. Then by Lemma C.1, the losses $\widetilde{\ell}_j$ are convex, nonnegative and $\zeta$-smooth.

**Lemma C.3.** *For $0 < \eta \leq 1/(2\zeta)$ and for any $w^* \in \mathbb{R}^d$,*

$$\sum_{j=1}^{m} \widetilde{\ell}_j(w_j) - \widetilde{\ell}_j(w^*) \leq \frac{\|w^*\|^2}{\eta} + 2\zeta\eta \sum_{j=1}^{m} \widetilde{\ell}_j(w^*) \ .$$

*Proof.* Fix any $w^* \in \mathbb{R}^d$. By a standard regret bound for online gradient descent with convex losses $\tilde{\ell}_j$ (e.g., Shalev-Shwartz & Ben-David, 2014; Hazan et al., 2016), we have

$$\sum_{j=1}^{m} \big(\widetilde{\ell}_j(w_j) - \widetilde{\ell}_j(w^*)\big) \leq \frac{\|w^*\|^2}{2\eta} + \frac{\eta}{2} \sum_{j=1}^{m} \|\nabla\widetilde{\ell}_j(w_j)\|^2 \ .$$

Since $\ell_j$ is nonnegative and $\zeta$-smooth, it follows that $\|\nabla\widetilde{\ell}_j(w_t)\|^2 \le 2\zeta\widetilde{\ell}_j(w_j)$. Thus,

$$\sum_{j=1}^{m}\big(\widetilde{\ell}_j(w_j) - \widetilde{\ell}_j(w^*)\big) \le \frac{\|w^*\|^2}{2\eta} + \zeta\eta\sum_{j=1}^{m}\big(\widetilde{\ell}_j(w_j) - \widetilde{\ell}_j(w^*)\big) + \zeta\eta\sum_{j=1}^{m}\widetilde{\ell}_j(w^*).$$

Rearranging terms, we obtain

$$\sum_{j=1}^{m}\big(\widetilde{\ell}_j(w_j) - \widetilde{\ell}_j(w^*)\big) \le \frac{\|w^*\|^2}{2\eta(1 - \zeta\eta)} + \frac{\zeta\eta}{1 - \zeta\eta}\sum_{j=1}^{m}\widetilde{\ell}_j(w^*)\,.$$

We conclude the proof by using $\eta\zeta \le \frac{1}{2}$ to upper bound the right-hand side. $\qquad\square$

We are now ready to prove Theorem 5.1.

*Proof of Theorem 5.1.* From Lemma C.3 we have

$$\sum_{j=1}^{m}\mathbb{E}[\widetilde{\ell}_j(w_j) - \widetilde{\ell}_j(w^*)] \le \frac{\rho_w^2}{\eta} + 2\zeta\eta\sum_{j=1}^{m}\mathbb{E}[\widetilde{\ell}_j(w^*)].$$

On the other hand, from Lemma C.2, we know that

$$\mathbb{E}[\widetilde{\ell}_j(w_j)] \ge \mathbb{E}[\mathcal{L}(w_j)] - 9k(\rho_w\rho_x + \rho_y)^2\delta$$

(since $w_j$ is independent of the $j$'th bag) and

$$\mathbb{E}[\widetilde{\ell}_j(w^*)] \le \mathcal{L}(w^*) \le L^\star.$$

Putting together, dividing through by $m$, and using the convexity of $\mathcal{L}(\cdot)$ we obtain

$$\mathbb{E}[\mathcal{L}(\bar{w}_m)] - \mathcal{L}(w^*) \le \mathbb{E}\left[\frac{1}{m}\sum_{j=1}^{m}\mathcal{L}(w_j) - \mathcal{L}(w^*)\right]$$

$$\le 9k(\rho_w\rho_x + \rho_y)^2\delta + \frac{\rho_w^2}{\eta m} + 2\zeta\eta L^\star.$$

Optimizing the bound with respect to $\eta \in (0, \frac{1}{2\zeta}]$, we can obtain

$$\mathbb{E}[\mathcal{L}(\bar{w}_m)] - \mathcal{L}(w^*) \le 9k(\rho_w\rho_x + \rho_y)^2\delta + \frac{4\zeta\rho_w^2}{m} + \sqrt{\frac{8\zeta\rho_w^2 L^\star}{m}},$$

for a choice of stepsize $\eta = \min\{\rho_w/\sqrt{2\zeta m L^\star}, 1/(2\zeta)\}$. (This is shown by considering two cases, according to whether $L^\star \le 2\zeta\rho_w^2/m$ or not.) To conclude, recalling that $\theta = \sqrt{8\log(1/\delta)/k}$ and

$$\zeta = (k\theta^2 + 1)\rho_x^2 = (8\log(1/\delta) + 1)\rho_x^2 \le 10\rho_x^2\log(1/\delta)\,,$$

we have obtained the bound:

$$\mathbb{E}[\mathcal{L}(\bar{w}_m)] - \mathcal{L}(w^*) \le 9k(\rho_w\rho_x + \rho_y)^2\delta + \frac{40\rho_x^2\rho_w^2\log(1/\delta)}{m} + \sqrt{\frac{80L^\star\rho_x^2\rho_w^2\log(1/\delta)}{m}}\,.$$

Setting $\delta = \frac{\rho_w^2\rho_x^2}{9km(\rho_w\rho_x + \rho_y)^2}$, we obtain

$$\mathbb{E}[\mathcal{L}(\bar{w}_m)] - \mathcal{L}(w^*) = \widetilde{O}\left(\frac{\rho_x^2\rho_w^2}{m} + \sqrt{\frac{L^\star\rho_x^2\rho_w^2}{m}}\right)\,.$$

Equating the right-hand side to $\beta$, and solving for $m$ gives the claimed result. $\qquad\square$

# D. Experiment details and additional results

We include here further details about our experiments that have been omitted from the main body of the paper.

**Datasets.**   We use the following datasets:

*MNIST Even vs Odd:* MNIST is a digit classification dataset consisting of 60,000 training examples and 10,000 test examples. Each image in the data is a $28 \times 28$ grayscale image. We replace the original labels by the parity of the digit. That is, even digits have label 0, while odd digits have label 1. No other processing is performed on MNIST.

*CIFAR-10 Animal vs Machine:* CIFAR-10 is a dataset where the goal is to classify images into one of 10 categories consisting of 50,000 training examples and 10,000 test examples. Each image in the data is a $32 \times 32$ three channel image. We replace the original labels by whether they are an animal or machine. That is, the classes 'bird', 'cat', 'deer', 'dog', 'frog', and 'horse' are all mapped to label 0, while classes 'airplane', 'automobile', 'ship', and 'truck' are mapped to label 1. No other processing is performed on CIFAR-10.

*Higgs:* The Higgs dataset consists of Monte-Carlo simulated particle accelerator data, where the goal is to distinguish between processes that create Higgs bosons and that do not. Each example has 28 features consisting of raw simulated measurements and several human-designed higher level features. The Higgs dataset has 11,000,000 examples. We use the first 10,000 examples as test data, and the remaining examples as training data.

*Adult:* The Adult dataset is derived from the US Census and the goal is to predict whether an individual's annual income exceeds \$50k per year. The data contains 32,561 training examples and 16,281 test examples. Each example consists of 14 features: numerical features 'age', 'fnlwgt', 'education-num', 'capital-gain', 'captial-loss', 'hours-per-week' and categorical features 'workclass', 'education', 'marital-status', 'occupation', 'relationship', 'race', 'sex', and 'native-country'. We one-hot encode each categorical feature and shift and rescale each numerical feature so that its values fall in the interval $[0, 1]$.

**Models.**   We use the following models:

*CNN for MNIST and CIFAR-10:* For both MNIST and CIFAR-10, we use the following CNN architecture:

- A convolution layer with 32 filters, $3 \times 3$ kernel, and ReLU activation.

- A max pooling layer with $2 \times 2$ window and stride.

- A convolution layer with 64 filters, $3 \times 3$ kernel, and ReLU activation.

- A max pooling layer with $2 \times 2$ window and stride.

- A dropout layer with 50% drop rate.

- A fully connected layer with a single output value.

*Higgs Model:* We implement the same deep network model proposed by Baldi et al. (2014) (and also used by Busa-Fekete et al. (2023)), which consists of four fully connected hidden layers each with 300 units and relu activations, followed by a fully connected output layer with a single neuron.

*Adult Model:* We implement a simple neural network consisting of a single hidden layer with 32 neurons and a single output neuron.

**Aggregate Losses.**   Next we describe the aggregate losses used and several important implementation details.

*Ours:* We simply use the unclipped version of our proposed loss given in (2) together with the implementation details described below for estimating $\mathbb{E}[h(x)]$ and $p$.

*Li et al.:* We use the loss

$$\ell_{\mathrm{LI}}(h, \mathcal{B}, \alpha) = \frac{1}{k} \left( k\alpha - \sum_{x \in \mathcal{B}} h(x) \right)^2 - (k-1) \cdot (\mathbb{E}[h(x)] - p)^2,$$

together with the same implementation strategy for estimating $\mathbb{E}[h(x)]$ and $p$.

*VanillaSQ and VanillaCE:* For a bag $\mathcal{B}$ with label proportion $\alpha$, let $\hat{\alpha} = \frac{1}{k} \sum_{x \in \mathcal{B}} h(x)$ denote the model's average prediction on $\mathcal{B}$. Then VANILLASQ is $(\hat{\alpha} - \alpha)^2$, while VANILLACE is $\ell_{\text{ce}}(\hat{\alpha}, \alpha)$, where $\ell_{\text{ce}}(\hat{y}, y) = -y \log(\hat{y}) - (1 - y) \log(1 - \hat{y})$ is the binary cross-entropy loss.

*EasyLLP:* Easy LLP corresponds to the loss

$$\ell_{\text{EASYLLP}}(h, \mathcal{B}, \alpha) = \frac{1}{k} \sum_{x \in \mathcal{B}} \left( w_1 \cdot \ell_{ce}(h(x), 1) + w_0 \cdot \ell_{ce}(h(x), 0) \right),$$

where $w_1 = k\alpha - (k - 1)p$ and $w_0 = k(1 - \alpha) - (k - 1)(1 - p)$. We estimate $p$ in the same way as described below.

*Implementation Details:* The three aggregate losses OURS, LI ET AL., and EASYLLP all require access to $p = \mathbb{E}_{(x,y) \sim \mathcal{D}}[y]$. In all cases, we estimate $p$ by taking the average label on the complete training data (which is equivalent to averaging the bag label proportions). The losses OURS and LI ET AL. both involve the average prediction of the current model $h$, $\mathbb{E}[h(x)]$. Given a minibatch that contains several bags, when computing the loss estimates on bag $i$, we estimate $\mathbb{E}[h(x)]$ to be the model's average prediction on the other bags. In particular, this provides an unbiased estimate of $\mathbb{E}[h(x)]$ that is uncorrelated with the other terms in the loss definition, which is important since correlations could eliminate the variance reduction achieved by the loss.

**Additional Results.** Figure 3 shows our results on all datasets, including Adult.

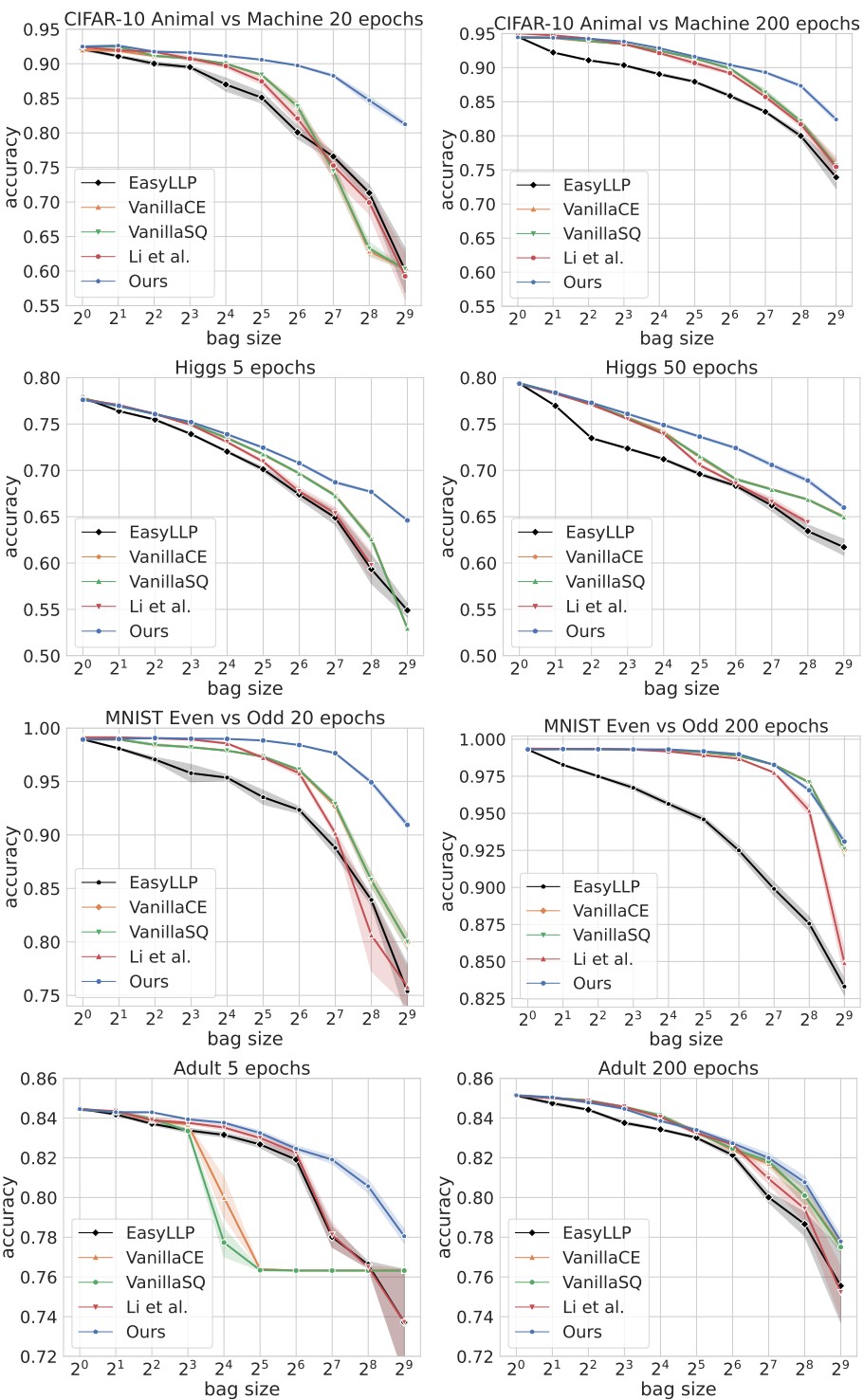

*Figure 3.* Plots showing the final test accuracy of models trained on each dataset as a function of the LLP bag size and number of training epochs. Error bars show one standard error in the mean over 10 repetitions. Each data point represents the accuracy achieved by a learning rate tuned for that bag size and loss.

