# OpenReview forum: "Nearly Optimal Sample Complexity for Learning with Label Proportions"
_ICML.cc/2025/Conference — ICML 2025 poster_

### Official Review · Reviewer_UVyu · 2025-03-09

**Overall Recommendation:** 3

**Summary:**

This paper investigates the problem of Learning with Label Proportions (LLP), where training data is provided in groups (or bags) rather than individual instances, and only aggregate label proportions are available for each group. The goal is to infer the original instance-level labels using the provided proportion information. To tackle this problem, the paper adopts a statistical learning perspective and aims to minimize the excess risk under both realizable and non-realizable settings. The authors focus on empirical risk minimization (ERM) and stochastic gradient descent (SGD) and propose a variance reduction technique that theoretically achieves a tighter regret bound, improving the learning performance in LLP. The effectiveness of the proposed method is evaluated through experiments on CIFAR-10, MNIST, and Higgs5 datasets, demonstrating its superiority over existing baseline approaches.

**Claims And Evidence:**

Most of the claims are provided with evidence.

**Essential References Not Discussed:**

No additional references needed.

**Experimental Designs Or Analyses:**

No.

**Methods And Evaluation Criteria:**

No.

**Other Comments Or Suggestions:**

Please check the weaknesses.

**Other Strengths And Weaknesses:**

Strengths
- Theoretical Contributions:
  - The paper establishes solid theoretical guarantees for LLP, particularly by addressing the necessity and sufficiency of hypothesis bias in LLP learning.
  - The theoretical results provide valuable insights into how hypothesis bias impacts learning performance in LLP settings, which is crucial for both model design and practical implementation.
- Generalizability of the Framework:
  - The proposed approach is not restricted to a specific learning paradigm but is applicable to both empirical risk minimization (ERM) and stochastic gradient descent (SGD).
  - This generalizability enhances the relevance and applicability of the proposed variance reduction technique, making it adaptable to different optimization and learning strategies.
- Empirical Effectiveness:
  - The experimental results show significant improvements over baseline methods, indicating the practical viability of the proposed approach.
  - The study evaluates the method on diverse datasets, including CIFAR-10 (image classification), MNIST (digit recognition), and Higgs5 (scientific data analysis), demonstrating its broad applicability.


Weaknesses and Areas for Improvement
- Clarity and Writing Quality:
  - The paper’s writing could be improved, as there are several typos and unclear descriptions that affect readability.
  - Some key arguments, especially regarding the theoretical framework, could be more clearly articulated.
- Motivation for the Proposed Learning Framework:
  - The paper lacks a clear motivation for why this particular learning framework is necessary.
  - While the authors demonstrate performance improvements, it remains unclear what specific challenges in existing LLP methods necessitate this new approach.
  - A more detailed discussion on how the proposed variance reduction technique uniquely addresses these challenges would strengthen the motivation.
- Justification for the Choice of ERM and SGD:
  - The authors consider both ERM and SGD as realizations of the LLP problem but do not sufficiently justify this choice.
  - ERM is a learning principle, whereas SGD is an optimization algorithm—this distinction should be discussed more explicitly.
  - It is unclear why the proposed variance reduction technique is equally applicable to both. Providing additional justification or empirical comparisons between different optimization techniques would help clarify this choice.
- Practical and Real-World Relevance:
  - The realistic implications of this study are not fully explored.
  - While the theoretical and empirical results are promising, the paper does not clearly articulate how LLP is applied in real-world scenarios.
  - Including concrete applications (e.g., medical diagnosis, fraud detection, or social science studies where label proportions are more accessible than instance-level labels) would strengthen the case for the practical impact of the work.

**Questions For Authors:**

If the authors can address my concerns, I will reconsider the score.

**Relation To Broader Scientific Literature:**

Theoretical contribution on a new framework for tackling LLP.

**Theoretical Claims:**

I have roughly read the proofs.

---

> ### Author Rebuttal · Authors · 2025-03-31
>
> - “... several typos and unclear descriptions… ”.
>
> We would appreciate pointers to typos, unclear descriptions, and key arguments that the reviewer found unclear so that we can improve them. Thank you!
>
> - On Motivation.
>
> As discussed in the right column of lines 23 – 32, a key modern motivation for the learning from label proportions framework is training conversion models for online advertising. Advertisers (or AdTechs working on their behalf) train models for predicting whether a user will convert (i.e., sign up for an account, buy a product, etc) if they click an advertisement. Since the ad click and conversion happen on different websites, collecting training data requires linking user behavior across the two sites. Historically, third party cookies or link decoration has made this straightforward. However, as web browsers move towards improving user privacy, there are several proposed APIs that allow for collecting cross-site data *in aggregate*. Many of these APIs can be used to directly measure the label proportion within a bag of examples. We view LLP as a principled technique for making the best use of these APIs for training conversion models. We will add more language along these lines to the introduction.
>
> - On challenges.
>
> The key challenge that we seek to address is to characterize the sample complexity of LLP: how much more data is required to train a model in the LLP framework than when all labels in the training set are available in the clear. We answer this question for the case of squared loss binary classification by giving the first (nearly) matching upper and lower bounds on the sample complexity of LLP. Now, given the fact that our loss estimate is almost unbiased (unlike those in Busa-Fekete et al, and Li et al.), the way in which the sample complexity can be improved over existing works is to improve the way the loss estimates concentrate at their expectation while keeping bias under control. It is intuitively clear that any method that selects a model by minimizing the estimated losses will be more reliable when the variance of this estimate is reduced (if the bias is also small). Our analysis leverages the lower variance of our estimates to derive improved sample complexity and convergence rates that are not possible with the methods of Busa-Fekete et al, and Li et al. Also, note that the theoretical best estimator is not even the one alluded to in Thm 4.1, but the *pairwise* (but not very practical) estimator of Thm 3.1, which is again a novel contribution of this paper. We will add text to the paper in order to better clarify these aspects.
>
> - Justification of ERM and SGD + “It is unclear why the proposed variance reduction technique is equally applicable to both.”
>
> We do not fully understand these comments. We focus on ERM and SGD because these are the most prominent learning paradigms. ERM is a learning principle – true – but the interpretation of our results should be rather clear: Any algorithm that performs ERM and operates with the loss function we designed for LLP enjoys the sample complexity guarantees we claim for ERM in Thm 4.1. Our results for ERM deal with the information-theoretic limits of LLP independently of increased computational difficulty in the underlying optimization problem. Our results for SGD study the impact of LLP on both the sample complexity and convergence rate of SGD for the case of linear hypotheses. In both cases, it is the design of an ad hoc loss function that enables improved results, not the learning algorithm itself. There are, on the other hand, some nuances, as this loss assumes prior knowledge of quantities that are usually unknown, and can only be estimated from data. We provide one such solution in Appendix B.2 in connection to ERM, and claim that similar fixes are possible for Sect. 3 (the pairwise estimator) and Sect. 5 (the SGD algorithm).
>
> - Practical and Real-World Relevance.
>
> As discussed above, we are mainly motivated by an extremely practical problem: training advertising conversion models from aggregate data collected via new privacy-preserving browser APIs. Improvements to the sample complexity of LLP in this context can directly lead to improved model performance. We will describe this more clearly in the introduction of the paper. As for experiments related to such data, this is a legitimate point that was also made by Reviewer YpnD. Unfortunately, most modern large-scale aggregated datasets (originating from online advertising) are proprietary. That led us to simulated aggregation environments on standard benchmark datasets. Yet, we believe our experimental environment does not make our experiments unrealistic.

---

> > ### Comment · Reviewer_UVyu · 2025-04-04
> >
> > Thanks for addressing my concerns; some of the replies helped me understand better. Hence, I am willing to raise my score but still stay borderline.

---

### Official Review · Reviewer_2x73 · 2025-03-12

**Overall Recommendation:** 3

**Summary:**

This paper provided the optimal sample complexity of LLP under square loss. Based on these analysis, the authors proposed an improved squared loss, which was reported to enjoy better classification accuracy on several binary classification datasets under LLP settings.

**Claims And Evidence:**

The claims made in the submission are generally supported by evidence and proofs.

**Essential References Not Discussed:**

An important seminal work [1] on multiclass LLP should be cited in the Introduction.

[1] Liu J, Wang B, Qi Z, Tian Y, Shi Y. Learning from Label Proportions with Generative Adversarial Networks. In NeurIPS 2019.

**Experimental Designs Or Analyses:**

Because the performance improvement of the proposed method only evident for big bag sizes, I think its contribution on LLP algorithm is limited.

**Methods And Evaluation Criteria:**

It is OK. However, due to the limitation to binary classification, many SoTA LLP solvers and dataset settings cannot be considered.

**Other Comments Or Suggestions:**

1. The variance comparison should be displayed in a more clearer way. (Maybe providing the quantitative results)
2. In fact, in some datasets, the improvement becomes significant under the largest bag size, i.e., $k=2^9$. It will be better to show the results under more other large bag sizes that the proposed method performs better than other existing methods.

**Other Strengths And Weaknesses:**

**Strengths:**

The performance improvement of LLP for big bag sizes is significant.

**Weaknesses:**

1.	As we already have several work on binary classification, the analysis on multi-class scenarios should be worth noting and desirable.
2.	The squared loss is not common in current work. The analysis of other losses (e.g., CE loss) can improve the practicability the this work.

**Questions For Authors:**

1. Can you discuss how your work can fit in the situation of varying bag size? (i.e., the bag size is not the same among the bags). Maybe a bounded interval for the bag size is a good assumption for further discussion.
2. In terms of hypothesis complexity, the discussion on linear predictor in Section 5 seems to be too restricted. How can you extend the results to more complex hypothesis space.

**Relation To Broader Scientific Literature:**

This work is a seminal exploration of improving LLP improvement under large bag sizes, which enjoys a promising focus in practice.

**Theoretical Claims:**

Yes, I have roughly reviewed the correctness of all theoretical proofs provided in the paper.

---

> ### Author Rebuttal · Authors · 2025-03-31
>
> - Extensions to Multi-class and Other Losses.
>
> We agree with the reviewer that extensions to the multiclass setting and losses other than the squared loss would be of interest. Having said that, our work is the first to provide matching upper and lower bounds on the sample complexity of learning from label proportions for binary classification with squared loss. In other words, if we go outside the (easy) noise-free realizable setting considered in Li et al. (2024), all prior work has suboptimal sample complexity even in this relatively simple setting. Our ongoing work explores extensions of our results to the multiclass setting and other losses. We expect the extension to multiclass to be relatively easy (but please see also the response to Reviewer Rv2q on this point). On the other hand, we feel that the extension to losses other than square loss will be more challenging (if we want to retain the minimum variance property of the estimators), thus belonging in a follow-up paper.
>
> - Essential References.
>
> Thank you for bringing to our attention the missed reference. We will include a citation in the introduction!
>
> - Varying Bag Size.
>
> This point was also raised by Reviewer Rv2q. Since our loss function produces an (almost) unbiased estimate of the per-example squared loss from a single bag, applying it to a collection of bags of varying sizes will still result in an overall (almost) unbiased estimate of the loss. And, compared to prior work, the variance of our per-bag estimates will all scale better with the size of the bags, so we expect to have lower variance and improved sample complexity even when the bags have varying sizes. Our restriction to fixed-size bags is only for clarity of presentation. We will add a comment to the paper on this point. As for the resulting sample complexity, please see the response to Reviewer Rv2q (“On uniform bag sizes”).
>
> - Variance Comparison.
>
> Following the reviewer’s suggestion, below is a simple scenario where the difference in variance between our estimator and Li et al. 's is readily assessable. Consider the simple setting when $x$ is uniform in $[0,1]$, $h^*(x) = x^2$, and $\hat h(x) = x$. We compute the label marginal $p$ exactly, and estimate $E[h(x)]$ in the same way as our learning experiments using SGD batches of size 1024 and a total of $n=2^{20}$ examples. From the table below, we see that the variance of our estimate stays roughly constant, while that of Li et al. grows as the bag size increases.
>
> |   bag size |      Ours |   Li et al. |
> |-----------:|----------:|------------:|
> |          2 | 0.0336042 |   0.0593875 |
> |          4 | 0.0363739 |   0.0931322 |
> |          8 | 0.037708  |   0.155778  |
> |         16 | 0.0385156 |   0.282495  |
> |         32 | 0.0392801 |   0.540059  |
> |         64 | 0.0414068 |   1.06511   |
> |        128 | 0.0478653 |   2.21354   |
> |        256 | 0.0565216 |   4.84059   |
>
> - Larger Bag Sizes:
>
> Following the reviewer’s suggestion, the following table reports final test accuracy for each method on each dataset using bags of size $k = 1024$. The number in the parenthesis is one standard error in the mean over 10 repetitions. The number of epochs is the same as the longer training run for each dataset, and we increased the SGD batch size to 2048 so that each batch contains more than one bag (which is necessary for our estimates of $E[h(x)]$). Otherwise, the experimental setup is identical to that of Section 6. On MNIST, Cifar-10, and Higgs, we see a large accuracy gap using our method compared to baselines, and on UCI Adult we slightly outperform all baselines, further supporting our theory.
>
> |           | mnist              | cifar10            | uci adult          | higgs              |
> |:----------|:-------------------|:-------------------|:-------------------|:-------------------|
> | Ours      | 0.876 (+/- 0.0047) | 0.796 (+/- 0.0115) | 0.770 (+/- 0.0024) | 0.628 (+/- 0.0013) |
> | Li et al. | 0.789 (+/- 0.0158) | 0.628 (+/- 0.0182) | 0.710 (+/- 0.0445) | 0.602 (+/- 0.0024) |
> | VanillaCE | 0.847 (+/- 0.0063) | 0.665 (+/- 0.0093) | 0.762 (+/- 0.0000) | 0.570 (+/- 0.0039) |
> | VanillaSQ | 0.848 (+/- 0.0066) | 0.663 (+/- 0.0084) | 0.762 (+/- 0.0000) | 0.568 (+/- 0.0041) |
> | EasyLLP   | 0.783 (+/- 0.0158) | 0.617 (+/- 0.0174) | 0.709 (+/- 0.0485) | 0.603 (+/- 0.0033) |
>
> - Hypothesis Complexity in Section 5 is too restrictive.
>
> We agree that restricting to linear hypotheses in Section 5 is restrictive. Still, remember that in order for SGD to have global convergence guarantees (which are the focus of this paper), the function
>
> $\qquad w  \rightarrow E[ \ell(h(x; w), y) ] $
>
> should be convex in $w$ (here $\ell$ is any convex loss function---in our case, the squared loss), and unless $h$ is a linear function in $w$, it is unlikely that this will be the case. And indeed, assuming a linear prediction rule is standard in convex, globally convergent analyses of SGD in Machine Learning contexts.

---

> > ### Comment · Reviewer_2x73 · 2025-04-05
> >
> > Thanks for the responses. The author has addressed most of my concerns and I am willing to raise my score to borderline acceptance.

---

### Official Review · Reviewer_Rv2q · 2025-03-13

**Overall Recommendation:** 5

**Summary:**

This paper addresses the challenge of Learning from Label Proportions (LLP) by analyzing the i.i.d. bag sampling setting. The authors provide improved sample complexity bounds in both realizable and non-realizable settings, using variance reduction methods, surpassing prior results. They further demonstrate the practical benefits of their approach through experiments, showing performance gains over baselines, especially in the critical regime of large bag sizes.

**Claims And Evidence:**

Yes.

**Essential References Not Discussed:**

N/A.

**Experimental Designs Or Analyses:**

Yes.

**Methods And Evaluation Criteria:**

Yes.

**Other Comments Or Suggestions:**

N/A.

**Other Strengths And Weaknesses:**

This paper is highly readable, contributes significantly to the field of LLPs, and presents novel results for the ICML community. I recommend acceptance, with the following suggestions for improvement.
1. The results rely on certain assumptions, such as the bag size assumption in Theorem 3.1.
2. The setting assumes a uniform bag size, which limits its practical applicability.
3. The results are limited to the squared loss. However, the square loss is often not ideal in practice. Exploring other loss functions, such as the log loss, would be more beneficial.
4. The observed gains compared to Li et al. (2024) are marginal for the majority of bag sizes. Could the authors discuss this observation?

**Questions For Authors:**

1. Could you please elaborate on the technical novelty and the challenges you faced when proving improved bounds, especially in comparison to Busa-Fekete et al. (2023) and Li et al. (2024)?

2. Could you elaborate on the challenges involved in generalizing your framework to multi-class classification?

**Relation To Broader Scientific Literature:**

The discussion of related work in Section 1.1 is appropriate.

**Theoretical Claims:**

Yes.

---

> ### Author Rebuttal · Authors · 2025-03-31
>
> - On uniform bag sizes.
>
> We present our results using a uniform bag size for simplicity, but in principle the loss estimate we propose here can be applied even in situations where we have bags of different sizes, and we should still expect to see improvements due to the reduced variance at each bag. The corresponding sample complexity will replace the dependence on the uniform bag size $k$ (see lines 262-271, right column) with a dependence on the average bag size $(k_1 + k_2 … + k_m)/m$, being $k_j$ the size of the $j$-th bag. We will add this observation to the paper.
>
> - On the use of square loss.
>
> We agree that extensions to other losses would be beneficial (and this is the focus of our ongoing work), but we found the squared loss setting already quite challenging.
>
> - Observed gains compared to Li et al. (2024) marginal for the majority of bag sizes.
>
> The variance of our loss estimates has a substantially improved dependence on the bag size compared to Li et al.’s loss estimate. We expect the performance of both loss estimates to be similar for small $k$ (when the variances are comparable), and for very large $k$ but fixed $n$ (when the bags are so large that the LLP algorithms simply do not have enough signal). But for large values of $k$, where learning is still possible, we expect our loss to perform better. Please see also the extra experiments we ran in response to Reviewer 2x73, where we show that the gap between our method and Li et al. 's widens substantially as $k$ grows (e.g., for $k$ = 1024).
>
> - On the technical novelty and the challenges in comparison to Busa-Fekete et al. (2023) and Li et al. (2024).
>
> The technical novelty is multi-faceted.
> 1. The main technical novelty is recognizing that the methods proposed in both Busa-Fekete et al. and Li et al. cannot provide the best sample complexity as a joint function of the regret bound $\beta$ and the bag size $k$ (this sample complexity is akin to a “privacy vs. utility” tradeoff – the bigger $k$ the more private the labels), unless something substantially different is tried in the way the estimators are designed. The variance reduction methods we proposed based on clipping and centered variables come at the price of making our estimators *biased*, and we need to keep both bias and variance under control. For square loss, we have shown that this is the best one can hope for (up to log factors).
> 2. Further, the fast rates claimed in Li et al. only apply to the easy situation where the best in class has *zero* loss (thereby ruling out the possibility of noise in the labels) – this is more restricted than the standard *realizable* scenario, where the Bayes optimal predictor coincides with the best in class hypothesis. This requires a more careful analysis than Li et al. 's when estimating the unknown parameters $E[h(x)]$ and $p$ (see Appendix B.2).
> 3. Moreover, we show that, in principle, there is an even better algorithm for LLP (even for square loss), which is based on pairwise comparisons and eliminations, described in Sect. 3 (for the two function case). That algorithm is better in that it does not have extra $\log k$ factors in its sample complexity bounds (at the cost of assuming $k$ large), but it is clearly not practical, since it requires doing comparisons for all pairs of functions in the hypothesis space. In passing, we also improve Li et al. ‘s lower bound by log factors (again, in the two hypothesis case).
> 4. Last but not least, we have shown that the sample complexity improvements are tangible, in that they are reflected in our experimental evidence.
>
> - Extension to multi-class classification.
>
> We do not expect the extension to multiclass to be hard. The main technical challenge is to come up with an estimator whose variance will depend on the number of classes *linearly* (or even sublinearly, if at all possible), instead of quadratically.

---

### Official Review · Reviewer_YpnD · 2025-03-19

**Overall Recommendation:** 3

**Summary:**

The paper presents a theoretical analysis for the problem of learning with label proportions (LLP) where training examples are grouped into “bags” and only the aggregate label proportion of each bag is observed (rather than individual labels). The authors propose a theoretical analysis of LLP under the square loss, proving nearly optimal sample complexity results. Algorithmically, the paper introduces carefully designed variants of standard learning methods (Empirical Risk Minimization and Stochastic Gradient Descent) equipped with variance-reduction techniques.

## update after rebuttal
Thanks to the authors for the clarifying answers. I am still inclined to keep the recommendation to weak accept as the paper is still largely suited for a journal publication given the it is largely based on the proofs in the appendix and requires an appropriate scrutiny.

**Claims And Evidence:**

Overall, the paper's claims seem generally reasonable. Some of the points as mentioned below are unclear :

Overall, the paper is quite dense, and understanding it is heavily dependent on having to read the appendix,  which contains long proofs. It is somewhat unreasonable to expect to read along with many other papers to review in the allocated time, with other constraints. In that sense, the paper should provide intuition of the main results in the main body, which it provides only partially, and hence the paper is largely suitable for a journal. On the other hand, I tried to check the claim of the scenario when \Delta and \beta are unknown (lines 190-195), and it seems there isn’t any proof in B.2.

Also, I am not sure about the claims in lines 228-232, how can one estimate E[h(x)] and E[h*(x)] from unlabeled data. Even in standard supervised setting, when h* is unknown, how could it be possible to make such estimates. In the given settings, when bag size becomes larger, this estimate would become worse in my opinion, so these assumptions seem quite strong.

The sample complexity result of realizable case O(k/beta) is not of much practical importance. While the other scenarios of non-realizability, the difference compared to previous results is not very significant O(k*k/\beta*\beta) in previous vs O(k/\beta*\beta) in this work. However, this matters only for large bag sizes, in which case the estimates of E[h(x)] and E[h*(x)] would become worse.

**Essential References Not Discussed:**

Not to my knowledge

**Experimental Designs Or Analyses:**

The experimental setup in the paper is generally reasonable and comparable to those used in previous related works mentioned in the paper and appropriate for evaluating an LLP method.

**Methods And Evaluation Criteria:**

The methods proposed in this paper are appropriate for the LLP problem and align with the theoretical claims. The authors build on standard learning algorithms – ERM and SGD – modifying them to handle bag-level supervision.

**Other Comments Or Suggestions:**

NA

**Other Strengths And Weaknesses:**

Strengths

1. Significance of Problem: The paper addresses an important problem setting (LLP) that is highly relevant for privacy-preserving machine learning.
2. Practical Algorithm and Empirical Validation: Unlike some theory-heavy papers, this work doesn’t stop at theory. The empirical results, showing improved accuracy with fewer samples, underscore that the theory has some practical merit.
3. Clarity and Organization: The paper is generally well-written and structured logically. The abstract clearly states the contributions, the introduction provides sufficient background and motivation (mentioning real-world scenarios which helps the reader care about LLP).


weakness

1. Incremental Aspects: A potential weakness is that the paper’s main theoretical claim, while valuable, is somewhat incremental with respect to very recent works Li et al. (2024), and Busa-Fekete et al. (2023).

2. Focus on Square Loss: The analysis is done under square loss, which is a convenient surrogate for classification. This is a limitation in the sense that the theoretical guarantees are for squared error/regret, not directly for 0-1 classification error. It’s possible to have a small square loss but still misclassify some fraction of points

3. Experimental Limitations: The empirical results are mostly on simulated LLP scenarios derived from standard datasets. The paper does not report results on an actual real-world aggregated-label dataset

4. Clarity in Technical Sections: While the paper is overall well-written, the theoretical sections are dense and may be challenging for readers not deeply versed in learning theory.

**Questions For Authors:**

As mentioned above

**Relation To Broader Scientific Literature:**

The paper seems to have a good discussion on relevant existing methods

**Theoretical Claims:**

Beyond the points raised above (in claims and evidence), the paper’s theoretical contributions seem reasonably sound. It would be helpful if the authors could help and clarify the above.

---

> ### Author Rebuttal · Authors · 2025-03-31
>
> - Claim in LL. 190-195 when $\Delta$ and $\beta$ are unknown + absence of proof in Appendix B.2.
>
> As quickly mentioned in L. 195-196, Appendix B.2 contains a proof of a similar statement that applies indeed to a harder situation (the one in Thm 4.1), where $E[h(x)]$ and $E[h^*(x)]$ are unknown. The idea here was just to point out that an argument similar to (but simpler than) the one contained in Appendix B.2 applies to the scenario of Thm 3.1. We will better clarify this  in the paper.
>
> - Claims in LL. 228-232 on how to estimate $E[h(x)]$ and $E[h^*(x)]$ from unlabeled data + “when bag size becomes larger, this estimate would become worse”.
>
> First, our claim contains a typo, and we apologize for that. Whereas $E[h(x)]$ only needs unlabeled data to be estimated, our estimate for $E[h^*(x)]$ makes use of the label proportions (that is, the aggregate labels). To estimate $E[h(x)]$, first observe that LLP does not hide the $x$ within each bag from us, so we have many i.i.d. feature vectors to evaluate $h$ on. This observation also applies to the estimation of $\Delta$ and $\beta$ in Thm 3.1. It is important to observe that the accuracy of these estimates only depends on the total number of samples $n$, rather than the total number of bags $m$. So, no, the estimate does not become worse when the bag size gets larger.
>
> To estimate $E[h^*(x)]$, we use the fact that for a bag with label proportion $\alpha$, we have that
>
> $\qquad E[\alpha] = E[(y_1 + … + y_k) / k] = E[(h^*(x_1) + … + h^*(x_k)] / k] = E[h^*(x)] = p .$
>
> So, by averaging the $\alpha$ values across many bags, we get an accurate estimate of $E[h^*(x)]$. Again, it is important to observe that this estimate does not degrade as the bag size $k$ gets larger, as its accuracy only depends on the number of samples $n$, instead of the number of bags $m$. Another way to see this is that the LLP data provides a collection of $m$ i.i.d. sums $\sum_i y_i$, each made up of $k$ terms. Now, an i.i.d. sum of i.i.d. variables just gives a bigger sum of i.i.d. variables, so we are again able to accurately estimate  $E[h^*(x)]$ with an accuracy that only depends on the total number of samples $n = m\times k$, instead of the total number of bags $m$.
>
> The way we made this work in Appendix B.2 is to simply separate the training set into two independent subsets, the first one for training and the second one for estimating $E[h(x)]$ and $E[h^*(x)]$. The accuracy at which $E[h(x)]$ and $E[h^*(x)]$ are estimated depending on $n$, not on $m$. So, even in this case, at a fixed $n$, the estimate does not become worse as the bag size gets larger. We will make all the above clearer in the main body of the paper.
>
> - On the significance of the bound improvements, $k^2/\beta^2$ vs. $k/\beta^2$, etc.
>
> We have to respectfully disagree on this point … . It is just the large bag size scenario that matters the most in practice. You may view this as akin to interplay between privacy (the bigger $k$ the more “private” the labels are) and utility (the regret bound $\beta$). Please see also the response to Reviewer UVyu about motivations.
>
> - On incrementality of the analysis.
>
> What we solved here was essentially stated as an open question by both Busa-Fekete et al. (2023), and Li et al. (2024) – see, e.g., the “Discussion” section in Li et al. This by itself should perhaps suggest that our work cannot be considered incremental . . .  . Technically, the sample complexity analysis is fairly different from Busa-Fekete et al., and closer to Li et al., but more involved, due to the effort to reduce the variance.
>
> - On the focus on square loss and the fact that squared error/regret does not directly translate into 0-1 classification error.
>
> Observe that we allow our models $h$ to have output in the interval $[0,1]$. As a special case, we can restrict the predictions of the models to be $0$ or $1$, making the square loss equivalent to the zero-one loss. Then our loss estimates allow one to estimate directly the zero-one loss of the model. We do not emphasize this in the paper because the zero-one loss is difficult to optimize computationally. But in fact, our Thm 4.1 about ERM seamlessly applies to 0/1 loss regret as well.
>
> - Empirical results mostly on simulated LLP scenarios.
>
> We agree that including datasets with natively aggregated labels would be interesting. Unfortunately, most modern large-scale aggregated datasets (originating from online advertising) are proprietary. Yet, we do not feel that simulating aggregation on standard benchmark datasets makes the experiments unrealistic.
>
> - On denseness and readability.
>
> We will work on improving the readability of the technical sections in the paper. Thank you!

---

### Decision · Program_Chairs · 2025-05-01

**Decision:**

Accept (poster)

**Comment:**

This paper presents a work about learning from label proportions (LLP), where only aggregated label information is available for groups of examples (bags), and the objective remains accurate prediction at the individual example level. The authors present new theoretical insights into the sample complexity of LLP under the square loss, which demonstrates bounds that are essentially optimal and that notably improve prior results. Algorithmically, the paper introduces refined variants of empirical risk minimization and stochastic gradient descent. The empirical results show that the proposed methods outperform existing baselines across several benchmark datasets. Overall, the paper makes both solid theoretical contributions and demonstrates practical effectiveness, which is meaningful in the study of LLP.

In the initial review phase, this paper received mixed recommendations, where the concerns were mainly reflected in the explanations/justifications of theoretical results and real-world relevance. After the rebuttal, the concerns in theory are addressed properly. The rationality of the simulation experiment has also been answered. All reviewers become positive about the work. AC recommends accepting it. The authors are encouraged to incorporate the reviewers’ constructive feedback in the final version to improve the clarity and impact of this work.